# Green fluorescent protein-like pigments optimise the internal light environment in symbiotic reef-building corals

Elena Bollati[1,2,3]*, Niclas H Lyndby[2,4], Cecilia D'Angelo[1], Michael Kühl[2,5], Jörg Wiedenmann[1], Daniel Wangpraseurt[6,7,8]*

[1]Coral Reef Laboratory, School of Ocean and Earth Science, University of Southampton, Southampton, United Kingdom; [2]Marine Biology Section, Department of Biology, University of Copenhagen, Helsingør, Denmark; [3]Department of Biological Sciences, National University of Singapore, Singapore, Singapore; [4]Laboratory for Biological Geochemistry, School of Architecture, Civil and Environmental Engineering, Ecole Polytechnique Fédérale de Lausanne (EPFL), Lausanne, Switzerland; [5]Climate Change Cluster, University of Technology Sydney, Sydney, Australia; [6]Department of NanoEngineering, UC San Diego, San Diego, United States; [7]Marine Biological Research Division, Scripps Institution of Oceanography, UC San Diego, San Diego, United States; [8]Bioinspired Photonics Group, Department of Chemistry, University of Cambridge, Cambridge, United Kingdom

*For correspondence:
elena.bollati@bio.ku.dk (EB);
dwangpraseurt@eng.ucsd.edu
(DW)

**Competing interest:** The authors declare that no competing interests exist.

**Abstract** Pigments homologous to the green fluorescent protein (GFP) have been proposed to fine-tune the internal light microclimate of corals, facilitating photoacclimation of photosynthetic coral symbionts (Symbiodiniaceae) to life in different reef habitats and environmental conditions. However, direct measurements of the in vivo light conditions inside the coral tissue supporting this conclusion are lacking. Here, we quantified the intra-tissue spectral light environment of corals expressing GFP-like proteins from widely different light regimes. We focus on: (1) photoconvertible red fluorescent proteins (pcRFPs), thought to enhance photosynthesis in mesophotic habitats via wavelength conversion, and (2) chromoproteins (CPs), which provide photoprotection to the symbionts in shallow water via light absorption. Optical microsensor measurements indicated that both pigment groups strongly alter the coral intra-tissue light environment. Estimates derived from light spectra measured in pcRFP-containing corals showed that fluorescence emission can contribute to >50% of orange-red light available to the photosynthetic symbionts at mesophotic depths. We further show that upregulation of pink CPs in shallow-water corals during bleaching leads to a reduction of orange light by 10–20% compared to low-CP tissue. Thus, screening by CPs has an important role in mitigating the light-enhancing effect of coral tissue scattering and skeletal reflection during bleaching. Our results provide the first experimental quantification of the importance of GFP-like proteins in fine-tuning the light microclimate of corals during photoacclimation.

## Editor's evaluation

This work provides interesting insight into and hypotheses about the ecological roles of fluorescent proteins (like GFP) that many scientists use as tools for their experiments, but whose natural roles are little understood. The work provides evidence that they modify the light environment available to the organism (here a coral and its symbionts) to allow for photosynthesis even under low light conditions. Thus, the work provides an interesting mixture of insight to scientists of other disciplines

to understand what they are working with and to ecologists/marine biologists to better understand light usage in the oceans. The paper is very well written.

## Introduction

Tropical coral reefs are solar-powered oases of marine biodiversity, where scleractinian corals are responsible for building the reef structures that host over 30% of all named marine multicellular species (*Fisher et al., 2015*). Corals rely heavily on photosynthesis by their dinoflagellate endosymbionts (Symbiodiniaceae) to meet their metabolic requirements (*Muscatine, 1990*). The distribution of reef-building corals is therefore constrained by the availability of photosynthetically active radiation (PAR, 400–700 nm), which is in turn determined by physical factors such as latitude, seasonality, water depth, and the optical properties of the water body (*Kirk, 1994*; *Kleypas et al., 1999*; *López-Londoño et al., 2021*). Variability in these parameters affects both the quantity and spectral quality of irradiance, exposing corals to a large range of light environments even across small spatial scales (*Anthony and Hoegh-Guldberg, 2003*; *Brakel, 1979*; *Eyal et al., 2015*). On shallow reef flats in clear water regions, corals are routinely exposed to excess irradiance that supersaturates symbiont photosynthesis (*Jimenez et al., 2012*; *Wangpraseurt et al., 2014b*), with short light flashes reaching over 3900 µmol photons $m^{-2}$ $s^{-1}$ due to the lensing effect of waves (*Veal et al., 2010*). Additionally, shallow-water corals are exposed to a broad irradiance spectrum covering UV to near infrared radiation (*Wangpraseurt et al., 2014b*). At the other extreme end of the scale, photosynthetic corals on mesophotic reefs at water depths between 50 and 150 m can receive as little as 1% of surface PAR, largely concentrated in the blue-green region of the spectrum and greatly reduced for UV and red wavebands (*Eyal et al., 2015*; *Lesser et al., 2009*; *Tamir et al., 2019*). Light conditions can also vary for individual colonies from the same water depth depending on colony morphology (*Kaniewska et al., 2011*; *Wangpraseurt et al., 2014b*) and the reef structure that surrounds them (*Falkowski and Dubinsky, 1981*; *Wangpraseurt et al., 2014b*).

Corals have evolved a number of mechanisms for optimising symbiont photosynthesis in the presence of spatial and temporal heterogeneity in light availability. Some of these mechanisms occur at the symbiont cell level, while others involve changes in the coral colony morphology and polyp behaviour (*Hoogenboom et al., 2008*; *Kahng et al., 2019*; *Kaniewska et al., 2013*; *Kaniewska and Sampayo, 2022*; *Kramer et al., 2021*; *Levy et al., 2003*; *Todd, 2008*). Coral symbionts can modify their light harvesting and energy quenching over timescales ranging from seconds to months. For example, non-photochemical quenching allows rapid dissipation of excess irradiance in the photosynthetic apparatus of symbionts, preventing photodamage in high light environments (*Einbinder et al., 2016*; *Gorbunov et al., 2001*; *Hoegh-Guldberg and Jones, 1999*).

Coral symbionts associated with deeper-water colonies have been frequently found to differ from their shallow-water counterparts in terms of photosystem II photochemical efficiency, maximum electron transport rates, and the organisation of light-harvesting antennae (*Einbinder et al., 2016*; *Hennige et al., 2008*; *Lohr et al., 2019*).

Photosynthetic pigment concentrations and symbiont cell numbers are dynamic within coral colonies, enabling them to acclimate to changes in irradiance; specifically, corals can increase their symbiont numbers as well as the per cell photosynthetic pigment content in response to a reduction in light levels and vice versa (*Eyal et al., 2019*; *Falkowski and Dubinsky, 1981*; *Martinez et al., 2020*; *Tamir et al., 2020*; *Titlyanov et al., 2001*). Symbiont community composition within colonies is also known to vary along light and temperature gradients, although the plasticity of this association and the extent to which it can contribute to photoacclimation appears to be species-specific (*Chan et al., 2009*; *Leydet and Hellberg, 2016*; *Einbinder et al., 2016*; *Kaniewska and Sampayo, 2022*; *Lesser et al., 2010*; *Martinez et al., 2020*; *Nir et al., 2011*; *Winters et al., 2009*). The interplay between light and temperature is also likely to play a role in shaping these communities on mesophotic reefs (*Kahng et al., 2019*; *Kahng and Kelley, 2007*), as the low temperatures sometimes encountered at depth affect the photosynthetic rates of symbiont species differently (*Grégoire et al., 2017*).

Coral host morphology is an important factor controlling the irradiance available for symbiont photosynthesis (*Einbinder et al., 2009*; *Helmuth et al., 1997*; *Kahng et al., 2019*; *Kahng et al., 2012*; *Kaniewska et al., 2011*; *Kaniewska and Sampayo, 2022*; *Nir et al., 2011*; *Ralph et al., 2002*; *Wangpraseurt et al., 2014b*). Coral colonies from low light environments maximise absorption

of downwelling light and assume horizontal plate-like morphologies that promote light collection, while shallow-water colonies assume self-shading morphologies, such as vertical plates or branches (*Anthony et al., 2005*; *Kaniewska et al., 2011*). Morphological adaptations make some species depth-specialists, such as the mesophotic coral *Leptoseris,* which can host photosynthetic symbionts down to depths of well over 100 m (*Kahng et al., 2020*; *Rouzé et al., 2021*). Depth-generalist corals on the other hand, such as *Montastraea cavernosa* in the Caribbean, thrive across light gradients of several orders of magnitude (*Lesser et al., 2010*). These corals often assume different morphologies in different light environments, allowing a single species to occupy a wider environmental niche. On mesophotic reefs, for instance, the depth-generalist corals *Paramontastrea peresi* and *Porites lobata* have growth forms that enhance light exposure, and their skeletons have higher reflectivity compared to their shallow water conspecifics, resulting in higher absorption by the symbionts (*Kramer et al., 2020*). Such morphological differences are partially due to phenotypic plasticity, as shown by transplantation experiments across depth gradients (*Muko et al., 2000*; *Willis, 1985*). However, the relationship between light and colony morphology is not always predictable (*Doszpot et al., 2019*) as other environmental variables such as hydrodynamics also affect colony morphology (*Soto et al., 2018*).

Light availability to the coral symbionts depends on the optical properties of the surrounding host tissue and the underlying skeleton (*Marcelino et al., 2013*; *Wangpraseurt et al., 2019*; *Wangpraseurt et al., 2012*). Coral tissue and skeleton scatter incident light with a strength and directionality that are highly variable across species and even between different structures within a single colony (*Enríquez et al., 2017*; *Marcelino et al., 2013*; *Wangpraseurt et al., 2019*; *Wangpraseurt et al., 2016a*). The interplay between tissue and skeleton scattering/reflection and algal absorption modulate coral photosynthesis (*Brodersen et al., 2014*; *Marcelino et al., 2013*; *Scheufen et al., 2017*; *Wangpraseurt et al., 2019*). This can lead to outstanding photosynthetic quantum efficiencies that are close to the theoretical maximum, at least for the coral species for which this parameter has been investigated (*Brodersen et al., 2014*).

Aside from variability in skeletal morphology, tissue-level changes such as contraction and expansion are known to affect light penetration (*Wangpraseurt et al., 2014a*; *Wangpraseurt et al., 2017b*). The thickness and composition of the tissue can be affected by a number of variables, including genotype, nutritional status, seasonality, and environmental history (*Harland et al., 1992*; *Jones et al., 2020*; *Leinbach et al., 2021*; *Lough and Barnes, 2000*; *Rocker et al., 2019*). Changes in tissue composition (e.g. lipids, collagen, and proteins) will affect the light-scattering properties of the bulk tissue (*Jacques, 2013*), although detailed investigations on coral optical properties in relation to tissue composition and nutrient status remain to be performed (*Lyndby et al., 2019*).

The coral host can also modulate incident light by synthesising host pigments. The most widespread coral host pigments belong to the green fluorescent protein (GFP)-like superfamily, a group of homologous pigments which are ubiquitous among the Cnidaria (*Alieva et al., 2008*; *Matz et al., 1999*; *Shimomura et al., 1962*). Pigments in this family share a similar structure based on a tripeptide chromophore capable of absorbing visible or UV radiation (*Nienhaus and Wiedenmann, 2009*). However, despite having similar structure, cnidarian GFP-like proteins exhibit great variability in their optical properties and regulation mechanisms, indicating that they are functionally diverse (*Aihara et al., 2019*; *Alieva et al., 2008*; *D'Angelo et al., 2008*; *Leutenegger et al., 2007*; *Nienhaus and Wiedenmann, 2009*; *Salih et al., 2000*; *Smith et al., 2013*).

GFP-like proteins with light-induced expression are found predominantly in corals from high-light environments, and they are transcriptionally regulated by blue-light intensity (*D'Angelo et al., 2008*; *Salih et al., 2000*). GFP-like proteins with light-induced expression include fluorescent proteins (FPs) with emission in the cyan, green, and red spectral regions (cyan fluorescent proteins [CFPs], GFPs, and red fluorescent proteins [RFPs]), which absorb light and re-emit it as red-shifted radiation, and non-fluorescent chromoproteins (CPs), which absorb light but have no or little fluorescence emission (*Alieva et al., 2008*; *D'Angelo et al., 2008*). These pigments accumulate in high concentration in colony parts with low symbiont densities, often areas of active growth, such as skeletal ridges, the tips of branching corals, the margins of plating corals, and areas affected by epibionts and mechanical damage (*D'Angelo et al., 2012a*; *Salih et al., 2000*). The increased photon fluxes inside the host tissue caused by the absence of light absorption by the symbionts, and light reflection and scattering by skeleton and host tissue components (*Enríquez et al., 2005*; *Wangpraseurt et al., 2017a*) have

been suggested to cause the high-level pigment expression in these colony parts (*Bollati et al., 2020*; *Smith et al., 2013*). Additionally, blue light-induced upregulation as part of an optical feedback loop has been shown to drive strong accumulation of these pigments in bleached corals during episodes of mild heat stress or in heat-free bleaching, caused, for instance, by nutrient stress (*Bollati et al., 2020*; *Wiedenmann et al., 2013*).

FPs and CPs with light-induced expression are usually localised ectodermally and have been shown to exert a photoprotective function on the underlying symbiont layer (*Gittins et al., 2015*; *Quick et al., 2018*; *Salih et al., 2000*; *Smith et al., 2013*). Upregulation of these pigments in growing parts of colonies and in bleached tissue has been proposed to promote (re)colonisation of host tissue by symbiont cells via the reduction of internal light stress in areas with low symbiont densities (*Bollati et al., 2020*; *D'Angelo et al., 2012a*; *Smith et al., 2013*).

GFP-like proteins are also widely distributed in corals that are depth generalists or even meso-photic specialists (*Eyal et al., 2015*; *Oswald et al., 2007*), where they are constitutively expressed, often at high levels, across a wide range of irradiance regimes (*Leutenegger et al., 2007*). It has long been proposed that fluorescent pigments in low light habitats could enhance coral photosynthesis by modulating the spectrum and intensity of light that reaches the symbiont cells (*Salih et al., 2000*; *Schlichter et al., 1994*). A light-enhancement mechanism has been suggested for a particular group of constitutive GFP-like proteins known as photoconvertible RFPs (pcRFPs), which are common in thick-tissue depth generalist and mesophotic coral species (*Ando et al., 2002*; *Eyal et al., 2015*; *Oswald et al., 2007*; *Wiedenmann et al., 2004*). pcRFPs emit green fluorescence when first synthe-sised, but their chromophores are irreversibly converted to a red-emitting species upon absorption of a near-UV photon ~390 nm (*Ando et al., 2002*; *Wiedenmann et al., 2004*). As coral GFP-like proteins are typically arranged in tetramers, near-UV-induced post-translational modification leads to the formation of tetramers that contain both green and red subunits; they can therefore absorb light in the blue-green range and re-emit it in the orange-red range via highly efficient Förster resonance energy transfer between adjacent subunits (*Bollati et al., 2017*; *Wiedenmann et al., 2004*). The orange-red waveband is scarce on mesophotic reefs, yet it can stimulate chlorophyll fluorescence deeper in the tissue compared to blue-green wavelengths because it is poorly absorbed by photosyn-thetic pigments, an effect analogous to green light-driven photosynthesis in plant leaves (*de Mooij et al., 2016*; *Smith et al., 2017*; *Vogelmann and Evans, 2002*). Indeed, pcRFP-containing corals have been shown to survive better under a simulated mesophotic light spectrum as compared to non-pigmented conspecifics (*Smith et al., 2017*).

While GFP-like proteins are likely to have additional functions for the coral holobiont (*Aihara et al., 2019*; *Ben-Zvi et al., 2022*; *Yamashita et al., 2021*), optimisation of the symbiont light environment is arguably one of their primary roles. The effects of GFP-like proteins on the light environment experi-enced by coral symbionts are not simply dependent on the spectral properties of the isolated protein; rather, they are a product of the environmental and physiological regulation of the protein within its three-dimensional optical context. For example, microsensor measurements and optical modelling have shown that scattering by GFP-like proteins can enhance light penetration at low GFP concentra-tions, while high concentrations of GFP increase backscattering and reduce light penetration through the coral tissue (*Lyndby et al., 2016*; *Taylor Parkins et al., 2021*). Variability in the distribution and arrangement of these proteins – diffuse or granular and ectodermal or endodermal – also affects how they interact with light (*Salih et al., 2000*; *Wangpraseurt et al., 2019*), further indicating that spec-trally similar GFP-like proteins can be involved in different light modulation mechanisms.

While compelling, the vast majority of evidence provided so far to support the role of GFP-like proteins in modulating the coral photosymbiont light environment during photoacclimation is indi-rect. This includes light measurements taken outside of the coral tissue (*Smith et al., 2017*; *Smith et al., 2013*), optical simulations (*Lyndby et al., 2016*), and experiments performed on coral-related animals (e.g. corallimorpharians) and isolated symbionts (*Smith et al., 2017*; *Smith et al., 2013*). So far, quantitative in vivo and in situ measurements of the symbiont light environment in response to light-driven changes in host pigment composition have been lacking. Here, we used advanced optical sensor technology to directly quantify how light-driven regulation of GFP-like proteins affects the intra-tissue light microenvironment of depth-generalist and shallow-water corals. We focused on the effects of (i) near-UV-driven photoconversion in corals containing pcRFPs, and (ii) blue light-driven upregulation of inducible CPs during bleaching and in active growth zones. Our study provides an

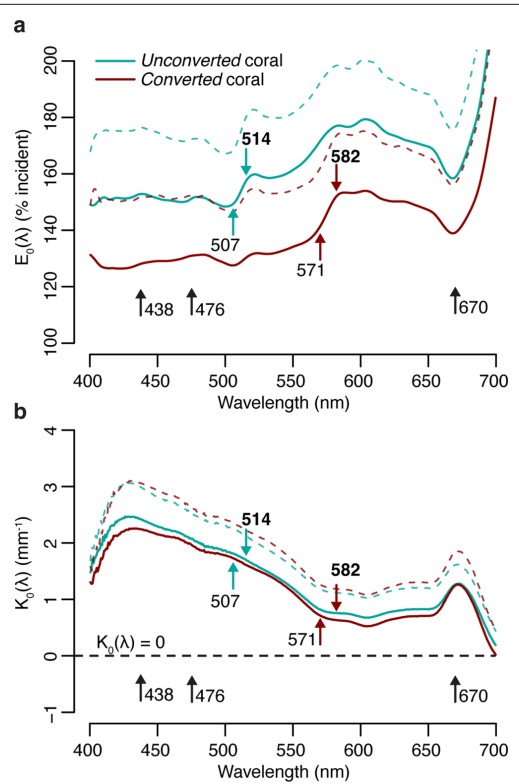

**Figure 1.** Spectral scalar irradiance under broad spectrum light in *M. cavernosa* before and after near-UV-induced photoconversion of the contained pcRFP. (**a**) $E_0(\lambda)$ at coenosarc surface, with excitation and emission peaks of unconverted and converted photoconvertible red fluorescent protein (pcRFP) highlighted (colourful arrows), as well as absorption peaks of Symbiodiniaceae photosynthetic pigments (black arrows). Mean (solid lines)+ SD (dashed lines); n=3 measurements on two fragments. (**b**) Spectral attenuation coefficient of $E_0(\lambda)$ from coenosarc surface to skeleton (average thickness = 560 µm). Mean (solid lines)+ SD (dashed line), n=3.

essential step towards understanding the role of these pigments in optimising the functioning of the coral-algal symbiosis under extreme light regimes.

## Results and discussion
### Photoconversion and pcRFPs in depth-generalist corals

Measurements of scalar irradiance taken at the tissue surface of *M. cavernosa* under white illumination showed an enhancement of 120–180% of the incident irradiance across the visible spectrum. This enhancement was detected in both the *unconverted* and *converted* (see Methods for definitions) tissue areas measured, with no significant differences between the two in any of the wavebands tested (t-test with Bonferroni adjustment, adjusted p>0.05; *Figure 1a*, *Appendix 1—table 1*). Vertical light attenuation was strongest between 410 and 470 nm (covering chlorophyll *a* absorption) and weakest in orange-red wavebands (*Figure 1b*).

The enhancement of scalar irradiance commonly observed at the surface of coral colonies is a result of light scattering by the underlying tissue and skeleton (*Kühl et al., 1995*; *Wangpraseurt et al., 2014a*), which displays great interspecific variability (*Marcelino et al., 2013*; *Wangpraseurt et al., 2019*). Depending on the scattering strength of the two compartments, light in deeper aboral tissue layers can be strongly reduced (*Wangpraseurt et al., 2012*) or enhanced (*Szabó et al., 2014*; *Wangpraseurt et al., 2020*), relative to the top (oral) tissue layers. Our measurements of the tissue attenuation coefficient for *M. cavernosa* indicate shading of the deeper tissue areas, particularly in the violet-blue range (*Figure 1b*). Additionally, previous measurements of optical scattering suggest that

light enhancement due to skeleton backscatter is limited for this species compared to other coral taxa (*Marcelino et al., 2013*). While these properties may confer bleaching resistance and provide photoprotection for the symbiont population in shallow water (*Marcelino et al., 2013*; *Wangpraseurt et al., 2012*), tissue shading greatly reduces carbon fixation rates in symbiont cells found in deeper tissue layers (*Wangpraseurt et al., 2016b*), which might negatively affect colonies in light-limited environments; yet, *M. cavernosa* is able to thrive at mesophotic depths (*Lesser et al., 2010*).

At the coral surface, spectral changes due to conversion of the pcRFP pool were observed at the absorption and emission peaks of the green (507 and 514 nm) and red chromophore (571 and 582 nm) (*Figure 1a*). Surface light enhancement by GFP-like proteins in corals can be attributed to a combination of elastic scattering (i.e. scattering without modifications of photon energy) and inelastic scattering (i.e. scattering with modification of photon energy, such as the case of absorption followed by fluorescence emission; *Lyndby et al., 2016*). Under white light, the degree of photoconversion of the pcRFP pool did not appear to significantly affect the spectral attenuation coefficient (*Figure 1b*), indicating that the effects of pcRFP scattering on attenuation of the incident are relatively small compared to the effects of tissue scattering and photosynthetic pigment absorption. Therefore it appears that,

for this pcRFP-containing coral specimen, host pigment-derived enhancement of light would be a minor component for the overall tissue light microclimate in a shallow-water environment, albeit with some variability due to water depth and the optical properties of the water column, as well as spectral differences between our white measuring light and sunlight. In contrast, corals on mesophotic reefs experience a much more narrow incident spectrum, where blue-green wavelengths that efficiently excite pcRFPs are abundant, while light in the orange-red spectral range is greatly reduced (*Eyal et al., 2015*; *Lesser et al., 2009*; *Tamir et al., 2019*). Under these conditions, the relative contributions of light scattering and fluorescence emission by pcRFPs are therefore likely very different from those measured under white light.

To isolate the contribution of fluorescence emission by the pcRFP pool to the tissue light microenvironment of *M. cavernosa*, we collected fluorescence emission profiles in the 505–700 nm range using a blue (455–505 nm) excitation light source (*Figure 2*, *Figure 2—figure supplement 1*). The peak emission wavelength of this light source matches (±3 nm) the irradiance maximum measured at 80 m depth in the Red Sea (*Eyal et al., 2015*), and over 50% of irradiance at 80 m falls within the 455–505 nm band (*Figure 2—figure supplement 1*). Therefore, while not a perfect match, this light source is spectrally more representative of mesophotic light conditions compared to the broad-spectrum LED (*Figure 2—figure supplement 1*). As a proxy for the degree of photoconversion, we used the ratio of 514:582 nm emission measured at the coral surface. This ratio was significantly higher in *unconverted* corals compared to those that had been exposed to near-UV light (*Figure 2—figure supplement 2*) for any duration (*partially converted* or *converted* corals, two-way ANOVA followed by Tukey's post-hoc test, adjusted p<0.01; *Figure 2*, *Appendix 1—table 2*). The ratio showed a significant decrease from surface to 350 µm tissue depth only for *unconverted* corals (adjusted p<0.01), and no significant differences were detected between the two groups at 350 µm tissue depth (adjusted p>0.05).

These results indicate that attenuation of light emitted by the unconverted chromophore is more pronounced relative to attenuation of the converted chromophore emission (*Figure 2a and c*). This is in agreement with the white light attenuation coefficient (*Figure 1b*), which was weaker in the orange-red range. The peridinin-chlorophyll *a*-protein light harvesting complex efficiently absorbs blue, green, and red wavelengths (*Jiang et al., 2012*), resulting in strong attenuation by symbionts in the oral gastrodermis (*Figure 1*); conversely, poor absorption of orange light by symbiont pigments allows this waveband to penetrate deeper into the tissue (*Figure 1*; *Lichtenberg et al., 2016*).

Using isolated symbionts and the sea anemone *Discosoma* as model systems for the symbiosis between cnidarian and Symbiodiniaceae, *Smith et al., 2017* showed that orange light has a higher potential to stimulate photosynthesis in symbiont cells found deeper in the tissue as compared to blue-green light. This effect is analogous to what is observed in plant leaves, where enhanced illumination with green light increases photosynthesis in deeper cell layers (*de Mooij et al., 2016*; *Vogelmann and Evans, 2002*). Therefore, based on this indirect evidence, it was proposed that orange fluorescence emission by pcRFPs could provide an energetic advantage for symbiotic corals in greater water depths dominated by blue light (*Smith et al., 2017*). Indeed, colour morphs with pcRFPs survived longer than non-pigmented conspecifics in simulated deep water light environments (*Smith et al., 2017*).

In this study, we aimed to directly measure the impact of pcRFPs on the *in hospite* light environment experienced by algal symbionts in coral tissue. We used the blue light excited fluorescence emission profiles from *M. cavernosa* (*Figure 2c*) at different stages of photoconversion (*Figure 3a–c*) to calculate the number of photons generated by fluorescence emission integrated over the green (*Figure 3d*) and red (*Figure 3e*) spectral range. Fluorescence emission by the pcRFP pool in the most converted state provided an amount of orange-red photons that ranged between 1 (close to the skeleton) and 5% (at the coral surface) of the incident blue irradiance, with intermediate values of around 3% at a tissue depth of 400 µm (*Figure 3e*). This direct measurement demonstrates that orange-red fluorescence from host pigments penetrates well in coral tissue. Furthermore, these values show that ectodermal pcRFPs can improve the illumination of deeper tissue layers exposed to the narrow mesophotic light spectrum. Thus, under a mesophotic light spectrum, the symbionts harboured by conspecific fluorescent morphs experience a spectrally different light environment depending on the pigment complement expressed by their host (*Ben-Zvi et al., 2021*). We obtained a similar result for a second species, *Echinophyllia* sp., by comparing specimens that contain a highly converted pcRFP to conspecific morphs that contain GFP but lack pcRFPs entirely (*Figure 3—figure supplement 1*).

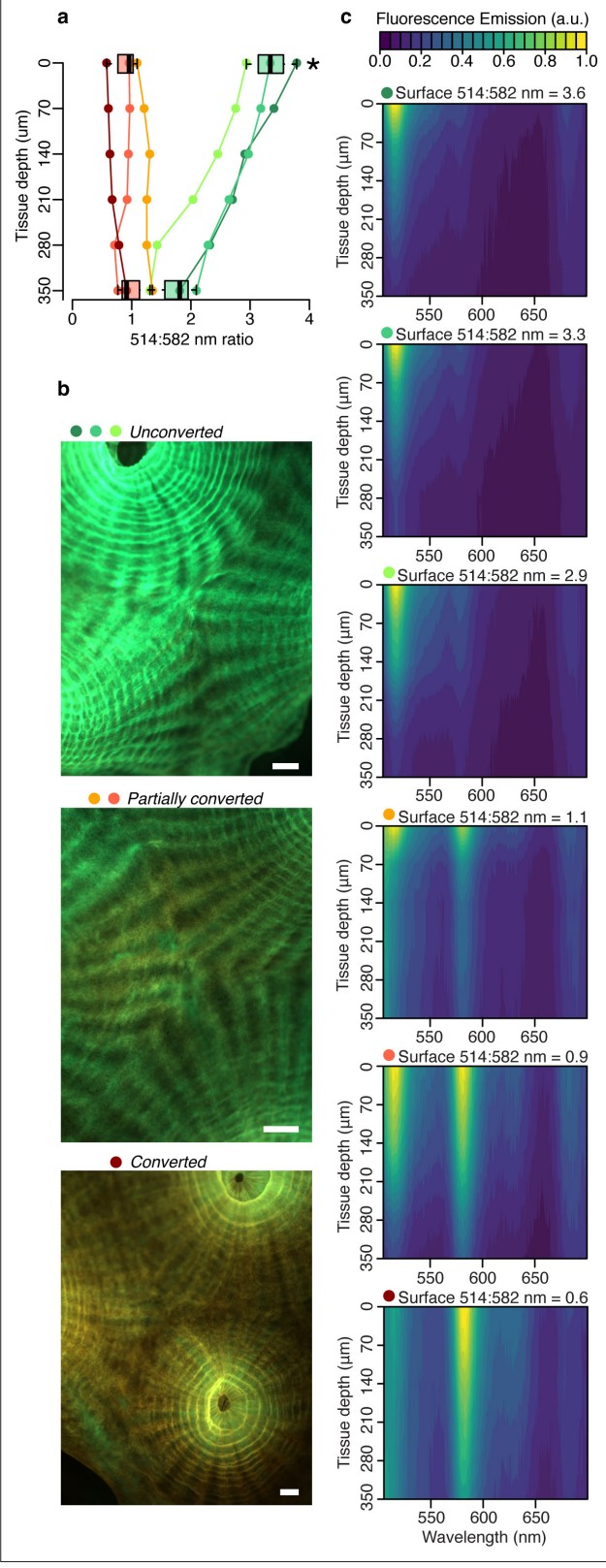

**Figure 2.** Fluorescence distribution in the tissue of *M. cavernosa* undergoing photoconversion. Scalar irradiance measurements collected under blue excitation 455–505 nm (***Figure 2—figure supplement 1c***). (**a**) 514:582 nm emission ratio, indicating the degree of photoconversion for each coral (*unconverted* >1, *partially converted* ~1, and *converted* <1). Boxplots show data for the surface and deepest data point, grouped as *unconverted* vs

*Figure 2 continued on next page*

*Figure 2 continued*

*partially converted* or *converted* (n=3 areas). Star represents adjusted p<0.05 in two-way ANOVA followed by Tukey post-hoc comparison. (**b**) Fluorescence micrographs of representative photoconversion stages. (**c**) Normalised fluorescence emission profiles (arbitrary units: a.u.). Scale = 1 mm.

The online version of this article includes the following figure supplement(s) for figure 2:

**Figure supplement 1.** The scalar irradiance measurement set up (**a**) and spectra (**b**) of the two light sources used for measurement ((**b**) KL-1600 LED; (**c**) xenon arc lamp + green fluorescent protein (GFP) Plus filter). In (**c**), dashed lines represent field measurements from 80 m depth in the Red Sea (*Eyal et al., 2015*), normalised to integrated irradiance.

**Figure supplement 2.** Spectra of light sources used for photoconversion (**a–c**) and bleaching (**d** and **e**) experiments (**a**, Lumiled Luxeon Rebel Green; **b**, xenon arc lamp +UV bandpass; **c**, Aquaray near UV; **d**, Lumiled Luxeon Rebel Deep Red; **e**, Lumiled Luxeon Rebel Royal Blue).

---

This confirms the relevance of our findings to other pcRFP-containing corals from mesophotic environments (*Eyal et al., 2015*).

To predict how spectral modifications by pcRFPs affect in situ light harvesting on mesophotic reefs, we used a published dataset of downwelling spectral irradiance from a coral reef in Eilat (*Eyal et al., 2015*; *Figure 4a*) and simulated the in vivo emission of orange-red light along a depth gradient from 10 to 130 m, using the *converted M. cavernosa* as a model (*Figure 4b*). The incident 455–505 nm light required to excite the pcRFP pool is relatively abundant down to 100 m water depth (1.5 μmol photos m$^{-2}$ s$^{-1}$); however, the amount of 560–650 nm light is only in the range of 1 μmol photons m$^{-2}$ s$^{-1}$ at 50 m and attenuates rapidly to about 0.1 μmol photons m$^{-2}$ s$^{-1}$ at 80 m (*Figure 4a*). As a consequence, the pcRFP pool between 50 and 80 m depth can effectively double the scalar irradiance in the 560–650 nm waveband compared to corals without pcRFPs (*Figure 4b*). On mesophotic reefs beyond 80 m water depth, simulated fluorescence emission by the pcRFP becomes the dominant source of

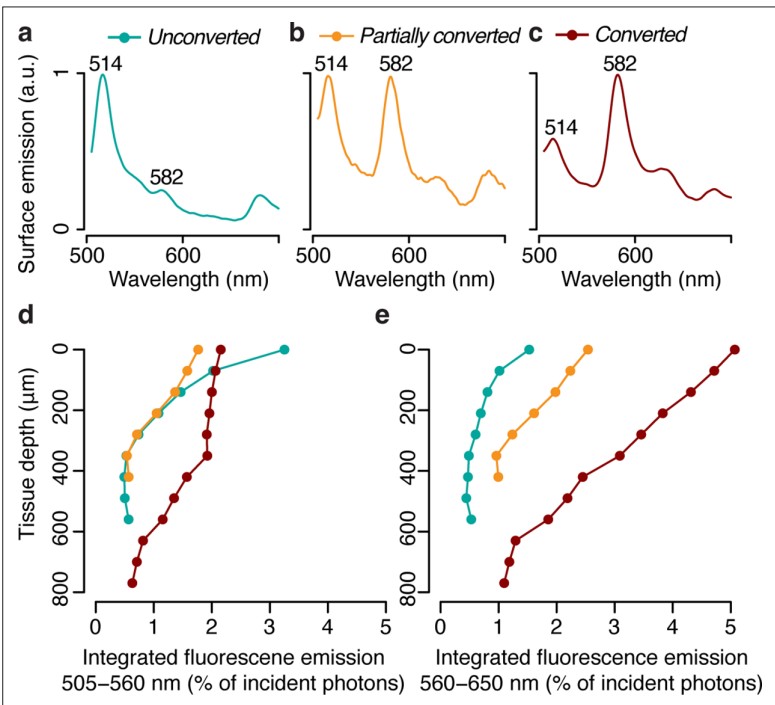

**Figure 3.** Photon scalar irradiance of fluorescence in the tissue of *M. cavernosa* at representative photoconversion stages. (**a**) Tissue surface emission spectra of *unconverted*, *partially converted*, and *converted* specimens (ex = 455–505 nm). (**b**) Integrated green (505–560 nm) and red (560–650 nm) fluorescence emission along a tissue depth profile, measured under blue excitation (455–505 nm, *Figure 2—figure supplement 1c*).

The online version of this article includes the following figure supplement(s) for figure 3:

**Figure supplement 1.** Photon scalar irradiance in the tissue of green and red morphs of *Echinophyllia*.

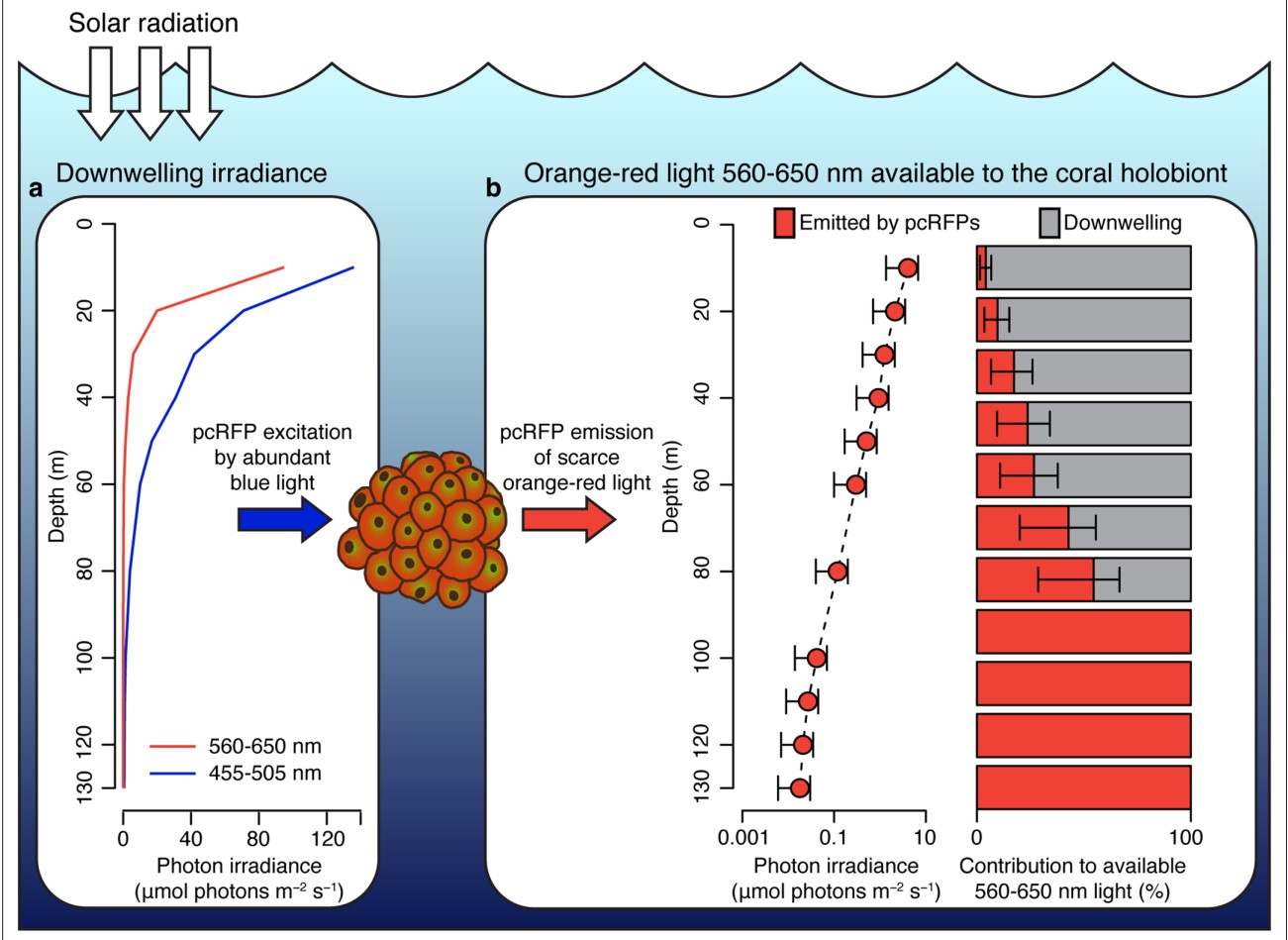

**Figure 4.** Spectral photon irradiance and wavelength conversion in corals with photo-convertible host pigments along a depth gradient. (**a**) Integrated photon irradiance in the 455–505 nm and 560–650 nm range measured on a mesophotic reef in Eilat, Red Sea (***Eyal et al., 2015***). (**b**) Measured 560–650 nm light generated by photoconvertible red fluorescent protein (pcRFP) emission and estimated relative contribution to the total amount of 560–650 nm light available to the coral holobiont. Error bars represent range between measurement at the coral skeleton (1%) and measurement at the coral surface (5%).

The online version of this article includes the following figure supplement(s) for figure 4:

**Figure supplement 1.** Relative contribution of fluorescence emission by photoconvertible red fluorescent protein (pcRFPs) and downwelling irradiance to the total 560–650 nm light available to pcRFP-containing corals along a depth gradient in the Bahamas (modelled irradiance data from ***Lesser et al., 2009***).

560–650 nm light for this *M. cavernosa* specimen. This calculation yielded similar results when applied to a dataset of modelled irradiance in the Bahamas (***Lesser et al., 2009***; ***Figure 4—figure supplement 1***), a natural habitat of this species, suggesting that our findings are relevant to mesophotic reef systems independent from potential differences in the underwater light environment.

We note that our analysis could lead to overestimation of predicted red fluorescence at shallow depths, where high light levels may theoretically excite the entire chromophore population, thus resulting in a non-linear relationship between incident light and fluorescence emission. However, the relationship between fluorescence emission and excitation light was linear up to at least 20 µmol photons m$^{-2}$ s$^{-1}$ of incident 455–505 nm light, the maximum intensity reached in our calibration measurements (Appendix 1, ***Appendix 1—figure 1***, ***Appendix 1—table 3***), corresponding to the incident irradiance (455–505 nm) measured at 48 m depth in the Red Sea (***Figure 4a***). Our calculations of fluorescence emission were thus considered accurate below 48 m water depth.

Even though this constitutes a dramatic spectral alteration in relative terms, it is important to note that (1) the absolute number of photons generated by pcRFP fluorescence is small compared

to what is provided by ambient light in shallow waters and (2) the *in hospite* irradiance is still dominated by the blue-green spectrum, which should be regarded as the main source of irradiance for coral symbionts in the mesophotic zone. Thus, under conditions of high solar radiation, the benefit of pcRFP fluorescence for photosynthesis may be negligible compared to the potential energetic investments related to maintaining a dense pcRFP pigment pool (*Leutenegger et al., 2007*; *Oswald et al., 2007*; *Quick et al., 2018*). However, under strong light-limitation (<40 μmol photons m$^{-2}$ s$^{-1}$), pcRFP-driven photosynthesis might be energetically advantageous, as suggested by the increased long-term survival of pcRFP-containing coral morphs compared to their brown conspecifics under mesophotic light regimes (*Smith et al., 2017*). Such small fitness advantage could have contributed as a driving factor in some of the multiple instances of convergent evolution of red-emitting homologues from green ancestor proteins (*Field et al., 2006*; *Gittins et al., 2015*; *Shagin et al., 2004*; *Ugalde et al., 2004*). Furthermore, the benefits and trade-offs associated with high level expression of pcRFPs may have contributed to the rise of colour morphs among individuals from genera such as *Trachyphyllia, Lobophyllia, Echinophyllia,* and *Montastraea* (*Field et al., 2006*; *Gittins et al., 2015*; *Oswald et al., 2007*) as a result of balancing selection as previously suggested for GFP-like proteins in both shallow and deep water corals (*Quick et al., 2018*).

It is important to note that host pigments are only one of many variables that influence the optical properties of coral colonies along a depth gradient. For *M. cavernosa*, one study reported differences in skeletal microstructure between colonies collected from different depths (*Beltrán-Torres and Carricart-Ganivet, 1993*). Another study revealed a shift towards heterotrophy and a change in symbiont community composition for the same species along a shallow to mesophotic gradient (*Lesser et al., 2010*). Genetic structuring of the *M. cavernosa* population with depth has also been reported (*Brazeau et al., 2013*). All these variables can potentially affect bio-optical properties and/or photosynthesis and thus affect the success of this species across steep light gradients. In this study, we controlled for these variables by choosing to perform our measurements on clonal replicates from long-term aquarium culture. This approach allows us to confidently attribute the measured changes in internal light environment to the changes in the pcRFP complement that we induced experimentally. Further studies will reveal if and how other variables may interact with pcRFPs functions to shape the symbiont light environment along natural depth gradients. Finally, intra-specific colour polymorphism of reef corals is frequently driven by different tissue concentrations of GFP-like proteins including CPs and pcRFPs (*Gittins et al., 2015*; *Quick et al., 2018*; *Smith et al., 2017*). Accordingly, representatives of different colour morphs can be expected to show deviating responses.

## Photoprotective CPs and bleaching in shallow-water corals

To evaluate the effect of high ectodermal concentrations of CPs on the light environment of coral symbionts *in hospite*, we performed spectral scalar irradiance measurements on colonies of *Pocillopora damicornis* and *Montipora foliosa*. Pink colour morphs of *P. damicornis* express a CP with absorption maximum at 565 nm (*Dove et al., 1995*), which can accumulate in high concentrations following bleaching (*Bollati et al., 2020*; *Figure 5a*). Brown morphs of this species do not produce detectable amounts of CP, although they contain a CFP (*Figure 5a*). The bare white skeleton of *P. damicornis* enhanced spectral scalar irradiance by 140–150% across the entire visible spectrum (*Figure 5b*). In brown morphs, the enhancement was reduced to <120% in the wavebands corresponding to the absorption peaks of symbiont photosynthetic pigments, while the enhancement remained close to 130% in the orange region of the spectrum (*Figure 5a*). Bleached corals expressing high pink CP concentrations experienced a surface light enhancement in the blue spectral region comparable to what was measured over bare skeleton; however, this was reduced to ~123% in the orange waveband corresponding to the absorption peak of the CP (*Figure 5b*). Measurements of the spectral attenuation coefficient from colony surface to skeleton showed that brown unbleached tissue is light-attenuating in the blue and red regions of photosynthetic pigment absorption but light-enhancing in the orange-red region; conversely, blue and red light are enhanced in bleached pink tissue, while orange light is attenuated (*Figure 5c*).

*M. foliosa* has several colour morphs, one of which produces high concentrations of a purple CP localised in areas of low symbiont density at the colony margins, while the inner area has low CP content and appears brown (*D'Angelo et al., 2012a*; *Figure 5d*). Measurements of surface spectral scalar irradiance on tissue found in the inner region of the colony showed similar spectral properties

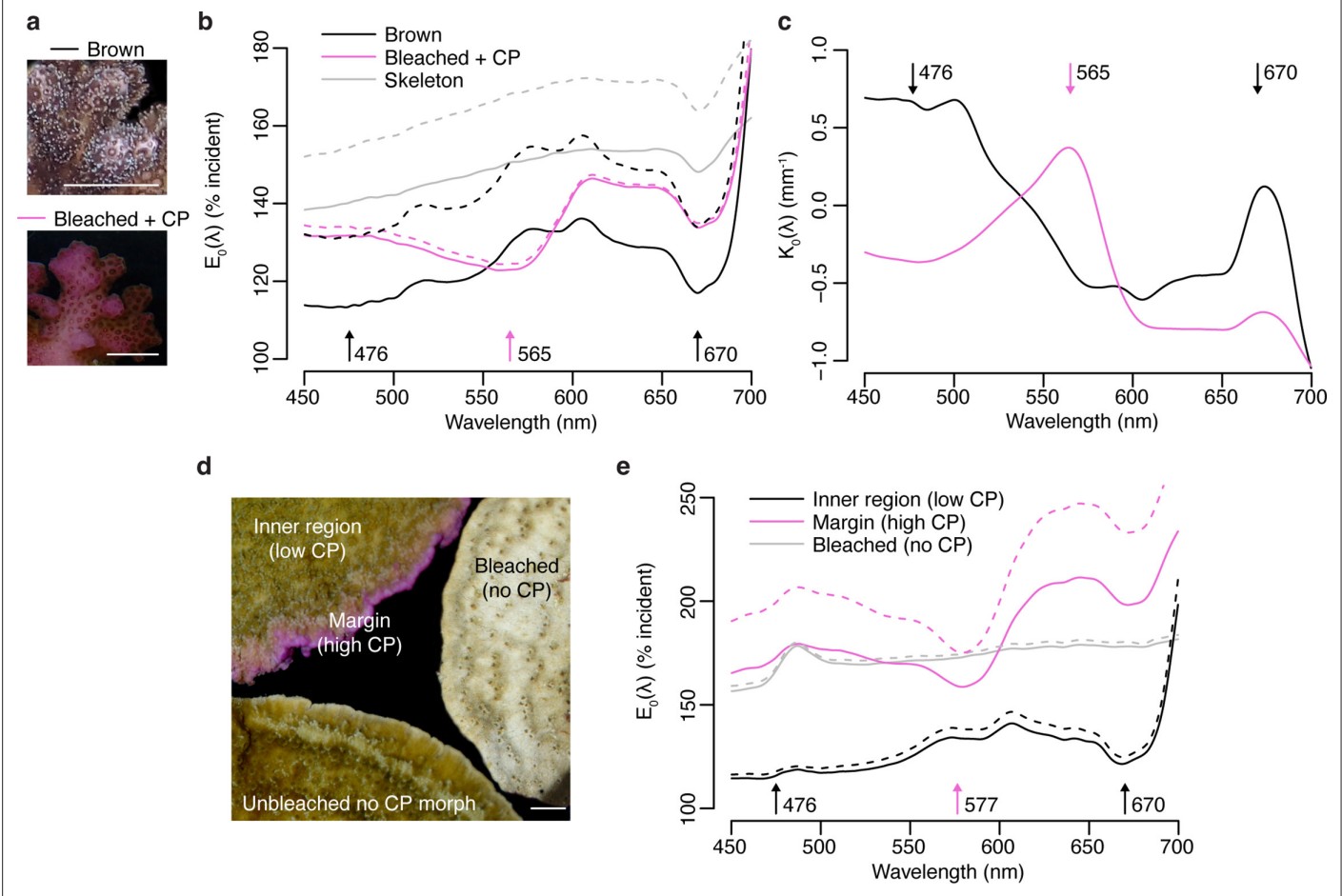

**Figure 5.** Effects of photoprotective chromoproteins (CP) on coral light environment. (**a**) Representative photographs of unbleached brown and bleached pink *P. damicornis*. (**b**) Spectral scalar irradiance ($E_0(\lambda)$) measurements at the coenosarc surface of unbleached brown *P. damicornis*, bleached *P. damicornis* with high CP content, and *P. damicornis* skeleton. (**c**) Spectral attenuation coefficient $K_0(\lambda)$ of scalar irradiance for unbleached brown and bleached pink *P. damicornis*, calculated from surface to skeleton (tissue thickness = ~300 μm). (**d**) Compilation of representative photographs of the unbleached high CP morph, unbleached no CP morph, and bleached no CP morph of *M. foliosa*. (**e**) Surface $E_0(\lambda)$ measurements of unbleached inner region (low CP), unbleached margin (high CP), and bleached no CP morph of *M. foliosa*. In (**a**) and (**d**), scale bar = 5 mm. In (**b**) and (**e**), mean (solid lines)+ SD (dashed lines), n=3 measurements. Arrows indicate peak absorption wavelength in nm for symbiont photosynthetic pigments (black) and CPs (pink).

to those of brown *P. damicornis* and exhibited 140% light enhancement, which was reduced to 110% in the symbiont photosynthetic pigment absorption wavebands (***Figure 5e***). In the high CP area at the colony margin, scalar irradiance showed a strong enhancement (>150%) comparable to what measured for bleached tissue in a CP-free morph (***Figure 5e***); however, this was heavily spectrally modified by the presence of the purple CP, which reduced light enhancement by up to 25% within its absorption spectrum compared to other spectral bands (***Figure 5e***). Screening by CPs via absorption was proposed as a photoprotective mechanisms in shallow-water corals by ***Smith et al., 2013***, who used external reflectance measurements as well as measurements on freshly isolated symbionts in seawater to provide indirect support for their hypothesis. Our data thus provide the first direct evidence that coral CPs do indeed reduce light penetration within their absorption spectrum, dramatically altering the *in hospite* light environment of coral symbionts.

The *P. damicornis* skeleton exhibits an intermediate level of microscale fractality and an intermediate scattering coefficient mass-fractal dimension ~1.75 and reduced scattering coefficient of ~7.5 mm⁻¹ (***Marcelino et al., 2013***), which implies a relatively efficient redistribution of light over the tissue (***Marcelino et al., 2013***). While this type of skeletal structure can help healthy corals optimise photosynthesis, during bleaching this can amplify the irradiance increase derived from the loss of

photosynthetic pigment absorption (*Enríquez et al., 2005*; *Marcelino et al., 2013*; *Wangpraseurt et al., 2017a*). Our data confirms that bleached *P. damicornis* tissue indeed experiences a strongly light-enhancing microenvironment, resulting from enhanced skeletal scattering due to reduced tissue absorption (*Wangpraseurt et al., 2017a*). Such strongly enhanced light microhabitat can also be found in healthy corals such as in the growing plate margins of *M. foliosa*, where the symbiont population is less abundant compared to the inner areas (*Figure 5d*). While blue light is attenuated in unbleached corals, the absence of symbionts in bleached corals or tissue areas with low symbiont densities (e.g. in areas of active tissue growth) results in the enhancement of blue light, leading to transcriptional upregulation of CPs (*D'Angelo et al., 2008*). This supports the notion that an optical feedback loop is the key mechanism underlying the colourful bleaching phenomenon (*Bollati et al., 2020*). Our data also show that high CP concentrations counteract the light-enhancing effects of bleaching within the CP absorption spectrum, as signified by the relative light enhancement measured in bleached corals in the orange region, a waveband known to result in light stress and bleaching for various coral species when provided at high irradiance (*Quick et al., 2018*; *Smith et al., 2013*; *Wijgerde et al., 2014*). Thus, optimisation of the light microenvironment by CPs in symbiont-depleted tissue could provide the necessary conditions for (re)colonisation by symbiont cells (*Bollati et al., 2020*; *D'Angelo et al., 2012a*; *Smith et al., 2013*).

## Orange-red light: a double-edged sword for coral holobionts

The data presented in this study clearly show that light-driven regulation of GFP-like host pigments can result in fine spectral tuning of the intra-tissue light field of corals. The pcRFPs and CPs examined here are characterised by very different optical properties, environmental regulation, and ecological distribution; however, they share a commonality in that both types of pigment regulate internal light fluxes in the orange-red spectral range. Enhancement of this waveband may be a double-edged sword for corals; on one hand, poor absorption by photosynthetic pigments allows better penetration (*Smith et al., 2017*), which can be exploited by pcRFP-containing corals to illuminate deeper tissue areas under light limitation. On the other hand, light amplification in symbiont-free tissue produces excess irradiance in the same waveband, which can cause photodamage unless counteracted by CP screening or other photoprotective mechanisms (*Quick et al., 2018*; *Smith et al., 2013*; *Wangpraseurt et al., 2017a*; *Wijgerde et al., 2014*). While our study has demonstrated the effects of host pigment regulation on the availability of orange-red light to the symbiont, the symbiont response to this spectral alteration remains to be demonstrated *in hospite* and quantified in terms of photosynthetic production. To date, very few studies have explored the effects of a red-shifted spectrum on symbiont photobiology *in hospite* (*Lichtenberg et al., 2016*; *Szabó et al., 2014*; *Wangpraseurt et al., 2014c*). The available data suggest that red illumination can support similar levels of photosynthetic production in deeper tissue areas compared to shallow ones, despite reduced light availability deeper in the tissue (*Lichtenberg et al., 2016*). Similar rates of oxygen evolution were also observed *in hospite* for symbionts exposed to either blue-shifted or red-shifted broad illumination; however, under low light conditions, red-shifted illumination resulted in lower electron transport rates (*Wangpraseurt et al., 2014c*). Finally, in cultured coral symbionts, the absorption cross-section of Photosystem II (PSII) is over 3× higher in the blue region compared to the orange-red; however, this difference becomes less pronounced when the symbionts are studied *in hospite* (*Szabó et al., 2014*). A more complete understanding of how coral symbionts acclimate to and utilise orange-red light *in hospite* is clearly needed in order to understand how spectral tuning of this waveband by host pigments impacts holobiont photosynthesis.

## Materials and methods
### Coral culture and aquarium set up

Coral colonies of *M. cavernosa* (one genet and three ramets), *Echinophyllia* sp. (two genets and one ramet of each), *P. damicornis* (two genets and one ramet of each), and *M. foliosa* (two genets and one ramet of each) were asexually propagated for >10 years in the experimental coral aquarium facility of the Coral Reef Laboratory, National Oceanography Centre, Southampton, UK. Each sample represents an autonomous colony with an individual, long-term growth history. Therefore, while genetically identical, these samples can be considered true biological replicates. These species and

colour morphs were selected for their established production of pcRFPs (*M. cavernosa* and *Echinophyllia* sp.) or photoprotective CPs (*P. damicornis* and *M. foliosa*) (*Bollati et al., 2020*; *D'Angelo et al., 2012a*; *Eyal et al., 2015*; *Oswald et al., 2007*). Additionally, their very long-term growth and propagation in our stable aquarium conditions ensured that these individuals were fully acclimated to the pre-experimental conditions, thus allowing to ascribe any observed patterns exclusively to the manipulated parameters – something that would not be straightforward for field collected or recently acquired colonies.

All corals were acclimated for >6 months to the light conditions specified in *Appendix 1—table 4*, at a temperature of ~25°C, salinity 31, and were fed twice weekly with rotifers. Nutrient levels were maintained around ~6.5 µM nitrate and ~0.3 µM phosphate (*Wiedenmann et al., 2013*; *D'Angelo and Wiedenmann, 2012b*) unless otherwise specified in the text and in *Appendix 1—table 4*. A detailed description of the aquarium set up including flow rates, volumes, co-cultured species, and monitoring of environmental parameters is presented in *D'Angelo and Wiedenmann, 2012b*.

## Spectral scalar irradiance measurements

Microscale spectral light measurements were conducted as described in *Wangpraseurt et al., 2017a* using scalar irradiance microsensors with a spherical tip diameter of 60 µm (*Rickelt et al., 2016*) and connected to a fibre-optic spectrometer (QE65000, Ocean Optics, USA). Microsensor profiles were performed from the coral skeleton upwards towards the tissue surface in increments of 20–100 µm using a motorised micromanipulator (Pyroscience GmbH, Germany) controlled by dedicated software (Profix, Pyroscience GmbH, Germany). The step size was adjusted according to the total coral tissue thickness with step sizes of 20 µm for *P. damicornis* (tissue thickness ~150–300 µm) and 70 µm for *M. cavernosa* (tissue thickness 400–800 µm) and *Echinophyllia* (tissue thickness 900 µm). For *M. foliosa*, only surface spectra were collected as the tissue was very thin (<60 µm). Scalar irradiance measurements were performed using vertical illumination from above, while the scalar irradiance microsensor was inserted into coral tissue at an angle of 45° relative to the incident light (*Figure 2—figure supplement 1a*). This measurement geometry was chosen to minimise artificial shading of the sensor tip while still allowing sensor penetration to the full tissue depth (*Rickelt et al., 2016*; *Wangpraseurt et al., 2012*). White LED illumination was provided by a fibre-optic lamp (KL-1600 LED, Schott GmbH, Germany; *Figure 2—figure supplement 1a,b*), producing 360–900 µmol photons m$^{-2}$ s$^{-1}$ at the measurement distance.

For *M. cavernosa* and *Echinophyllia*, we also measured the spatial distribution of emitted fluorescence. For this, narrow-waveband excitation in the blue spectral range was provided vertically incident by a xenon arc lamp equipped with a band-pass filter (Leica GFPplus filter, $\lambda$ =455–505 nm, *Figure 2—figure supplement 1c*), and scalar irradiance measurements in the 505–650 nm range were used to estimate fluorescence emission. The light source output (455–505 nm) at the measurement distance was 12–46 µmol photons m$^{-2}$ s$^{-1}$. No emission filter was used on the detector side as bleed-through of excitation light in this spectral range was minimal (<0.5%).

Scalar irradiance microsensor measurements were calibrated against a spectroradiometer (MSC15, Gigahertz-Optik, Germany) by measuring the incident downwelling irradiance from the white LED light source over a black light absorbing surface first with the microsensor in water, then with the spectroradiometer at the same distance from the light source in air (*Brodersen et al., 2014*; *Wangpraseurt et al., 2012*).

## Induction of photoconversion and measurements of pcRFP-containing corals

We performed scalar irradiance measurements in *M. cavernosa* at different stages of photoconversion to assess how near-UV-driven photoconversion of the pcRFP pool affects the symbiont light environment. We defined the stages of photoconversion by the relative proportions of green (514 nm) and red (582 nm) fluorescence emission, measured with 455–505 nm excitation at the colony surface. A 514:582 nm ratio >1 indicates a higher contribution of the green chromophore (defined as *unconverted* coral), a ratio ~1 indicates substantial contributions from both the green and the red chromophore (defined as *partially converted* coral), and a ratio <1 indicates a higher contribution of the red chromophore (defined as *converted* coral).

To obtain *unconverted* corals, we placed two colonies derived from fragmentation of the same mother colony under green incident irradiance provided by LED strips (Lumiled Luxeon Rebel Green, $\lambda$ =530 nm, and Full Width at Half Maximum (FWHM) = 60 nm; *Figure 2—figure supplement 2a*) for 40 days. This treatment resulted in the accumulation of the unconverted pcRFP (*Leutenegger et al., 2007*), evidenced by a 514:582 nm emission ratio >1. We then used one of these colonies to induce localised photoconversion by illuminating the coral tissue surface (spot diameter 3 mm) with a xenon arc lamp of a fluorescence microscope (Leica MZ10) equipped with a UV bandpass filter ($\lambda$ =355 nm and FWHM = 55 nm; *Figure 2—figure supplement 2b*) for 6 hr (*Eyal et al., 2015*); this waveband only partially overlaps with the action spectrum of pcRFP photoconversion (*Wiedenmann et al., 2004*) and is therefore less effective in inducing the photochemical reaction. In combination with the limited duration of the exposure, the near-UV treatment resulted in a mix of unconverted and converted pcRFP, as evidenced from the 514:582 nm ratio ~1 (*partially converted* coral). The second *M. cavernosa* colony was exposed to near-UV (NUV) irradiance provided by LED strips (Aquaray NUV, $\lambda$ =412 and FWHM = 18.5 nm; *Figure 2—figure supplement 2c*) for 10 hr; this treatment induces rapid photoconversion of the pcRFP pool (*Bollati et al., 2017*), resulting in a 514:582 nm emission ratio <1 (*converted* coral). All photoconversion experiments were performed in a flow-through system at a water temperature of 26°C regulated by a 25 W aquarium heater (Tunze).

We also assessed the contribution of fluorescence emission by pcRFPs to spectral scalar irradiance against other GFP-like proteins by performing measurements on two morphs of *Echinophyllia* sp. Both morphs had been kept under broad spectrum illumination long term, side-by-side in our mesocosm. The green morph contained a non-inducible GFP, while the red morph contained a CFP and a highly converted pcRFP (green:red emission ratio<<1; *Alieva et al., 2008*; *Bollati et al., 2017*; *Smith et al., 2017*).

## Experimental bleaching and measurements of corals containing inducible CPs

To test how bleaching affects irradiance in corals containing inducible CPs, we performed measurements on one healthy and one bleached individual each of *P. damicornis* and *M. foliosa* (three replicate measurements per sample per condition). A *P. damicornis* colour morph capable of expressing a pink CP with absorption maximum at 565 nm (*Bollati et al., 2020*; *D'Angelo et al., 2008*; *Dove et al., 1995*) was bleached by exposure to 400 µmol photons m$^{-2}$ s$^{-1}$ of red LED light (Lumiled Luxeon Rebel Deep Red, $\lambda$ =660 nm, and FWHM = 40 nm; *Figure 2—figure supplement 2d*) for 7 days. This treatment induces bleaching without inducing the expression of the light-sensitive genes responsible for CP production (*Bollati et al., 2020*; *D'Angelo et al., 2008*; *Smith et al., 2013*; *Wijgerde et al., 2014*). Bleaching was confirmed by visual comparison with >10 untreated specimens, which were kept in a separate area of the same aquarium under broad illumination from a metal halide lamp. Bleached fragments were then exposed to blue light (100 µmol photons m$^{-2}$ s$^{-1}$; Lumiled Luxeon Rebel Royal Blue, $\lambda$ =450 nm, and FWHM = 40 nm; *Figure 2—figure supplement 2e*) for 20 days to stimulate upregulation of the pink CP, resulting in a 'colourful bleaching' response (*Bollati et al., 2020*; *D'Angelo et al., 2008*; *Figure 2—figure supplement 2*). Alongside the bleached specimen, we performed spectral scalar irradiance measurements on (i) a *P. damicornis* skeleton and (ii) an unbleached brown morph of the same species, which expresses an inducible CFP but no CP.

For *M. foliosa*, we measured a morph capable of producing a CFP and a pink CP, with the latter accumulating in high concentrations at the colony margins (*Bollati et al., 2020*; *D'Angelo et al., 2012a*). We compared this healthy, unbleached coral with an *M. foliosa* morph that does not produce CPs, bleached by phosphate starvation (*Bollati et al., 2020*; *Rosset et al., 2017*; *Smith et al., 2013*) after 65 days in our high nitrate/low phosphate mesocosm (*D'Angelo and Wiedenmann, 2012b*). Both colonies were kept under broad spectrum illumination (200 µmol photons m$^{-2}$ s$^{-1}$; Aquamedic metal halide lamp).

## Data processing

All spectral data were corrected for spectrometer dark noise readings obtained under dark conditions and calibrated against a spectroradiometer as described above. For spectral measurements performed under incident white light, the spectra were normalised to the incident irradiance as described previously (*Wangpraseurt et al., 2012*).

The spectral attenuation coefficient of scalar irradiance $K_0(\lambda)$ was calculated from measurements obtained under white light following *Kühl, 2005*:

$$K_0\left(\lambda\right) = \frac{ln\left(\frac{E_0(\lambda)_1}{E_0(\lambda)_2}\right)}{z_2 - z_1}$$

where $E_0(\lambda)_1$ and $E_0(\lambda)_2$ are the spectral scalar irradiance measured at tissue depths $z_1$ and $z_2$. $K_0(\lambda)$ was calculated from surface to skeleton using the average tissue thickness (deduced from the mean number of equal steps performed by the micromanipulator from the skeleton to the upper tissue surface: 560 µm for *M. cavernosa* and 300 µm for *P. damicornis*).

For fluorescence emission measurements under blue excitation, integrated photon irradiance was calculated for different portions of the visible spectrum (incident: 455–505 nm; green emission: 505–560 nm; red emission: 560–650 nm) after converting spectral scalar irradiance to spectral photon scalar irradiance using Plancks equation (see *Lichtenberg et al., 2017*). Integrated photon irradiance over the same incident (455–505 nm) and red (560–650 nm) spectral ranges was also calculated for a dataset of field spectral scalar irradiance collected from Eilat, Israel (*Eyal et al., 2015*) and for a dataset of spectral scalar irradiance simulated from surface measurements collected in the Bahamas (*Lesser et al., 2009*). These values were used to calculate estimates of red fluorescence emission for the *converted M. cavernosa* at different water depths, by multiplying the integral of the 455–505 nm field radiance by the percentage of incident light re-emitted as 560–650 nm fluorescence shown in *Figure 3e*.

## Acknowledgements

This project was supported by the European Union's Horizon 2020 research and innovation programme (702911-BioMIC-FUEL, DW), the Gordon and Betty More Foundation (grant GBM 9325 to DW & grant GBMF9206 https://doi.org/10.37807/GBMF9206 to MK), and the Natural Environment Research Council (grant number NE/S003533/2 to JW and CDA).

## Additional information

### Funding

| Funder | Grant reference number | Author |
| --- | --- | --- |
| European Research Council | 702911-BioMIC-FUEL | Daniel Wangpraseurt |
| Gordon and Betty Moore Foundation | GMB 9325 | Daniel Wangpraseurt |
| Gordon and Betty Moore Foundation | GBMF9206 | Michael Kühl |
| Natural Environment Research Council | NE/S003533/2 | Jörg Wiedenmann |

The funders had no role in study design, data collection and interpretation, or the decision to submit the work for publication.

### Author contributions

Elena Bollati, Formal analysis, Investigation, Visualization, Writing - original draft; Niclas H Lyndby, Investigation, Writing – review and editing; Cecilia D'Angelo, Daniel Wangpraseurt, Conceptualization, Supervision, Investigation, Methodology, Writing – review and editing; Michael Kühl, Conceptualization, Resources, Supervision, Funding acquisition, Methodology, Writing – review and editing; Jörg Wiedenmann, Conceptualization, Resources, Supervision, Funding acquisition, Investigation, Methodology, Writing – review and editing

### Author ORCIDs

Elena Bollati ⓘ http://orcid.org/0000-0003-3536-4587

Niclas H Lyndby (iD) http://orcid.org/0000-0003-0533-9663
Michael Kühl (iD) http://orcid.org/0000-0002-1792-4790
Jörg Wiedenmann (iD) http://orcid.org/0000-0003-2128-2943
Daniel Wangpraseurt (iD) http://orcid.org/0000-0003-4834-8981

### Decision letter and Author response

Decision letter https://doi.org/10.7554/eLife.73521.sa1
Author response https://doi.org/10.7554/eLife.73521.sa2

---

## Additional files

### Supplementary files
• Transparent reporting form

### Data availability
Data used in this study is available from https://doi.org/10.5061/dryad.0gb5mkm1z.

The following dataset was generated:

| Author(s) | Year | Dataset title | Dataset URL | Database and Identifier |
|---|---|---|---|---|
| Bollati E | 2022 | Data from: Green fluorescent protein-like pigments optimize the internal light environment in symbiotic reef building corals | https://doi.org/10.5061/dryad.0gb5mkm1z | Dryad Digital Repository, 10.5061/dryad.0gb5mkm1z |

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

## Appendix 1

### Incident intensity calibration (supplementary methods)

To determine whether the relationship between excitation light intensity and emitted fluorescence was linear within our measurement range, we performed a calibration profile on a *converted Montastraea cavernosa* sample using a set of neutral density filters. At each depth, we collected fluorescence emission measurements under 455–505 nm excitation with five combinations of neutral density filters placed between the excitation light and the specimen. We then calculated the integrated photon scalar irradiance in the incident range (455–505 nm) at each tissue depth, the integrated green fluorescence emission (505–560 nm, *Appendix 1—figure 1a*), and the integrated red fluorescence emission (560–610 nm, *Appendix 1—figure 1b*). A linear regression was fitted for each tissue depth and each chromophore, with intercept set to zero (*Appendix 1—figure 1*); p values were corrected for multiple testing using the Bonferroni adjustment (*Appendix 1—table 3*).

**Appendix 1—table 1.** Results of t-test with Bonferroni adjustment for *Figure 1a*.

| Wavelength | t value | d.f. | p | Adjusted p |
|---|---|---|---|---|
| 438 | 1.572583 | 3.165662 | 0.209155 | 1 |
| 476 | 1.49222 | 3.538786 | 0.218869 | 1 |
| 507 | 1.638856 | 3.678655 | 0.182766 | 1 |
| 514 | 1.754119 | 3.500689 | 0.164409 | 0.986453 |
| 571 | 1.946208 | 3.84537 | 0.126343 | 0.758061 |
| 670 | 1.553197 | 3.589537 | 0.203304 | 1 |

**Appendix 1—table 2.** Results of two-way ANOVA followed by Tukey test post-hoc comparison for *Figure 2a*.

| Factor | d.f. | Sum Sq | Mean Sq | F | p |
|---|---|---|---|---|---|
| Photoconversion | 1 | 7.709 | 7.709 | 61.47 | <0.001 |
| Tissue depth | 1 | 1.646 | 1.646 | 13.13 | 0.007 |
| Photoconv:depth | 1 | 2.301 | 2.301 | 18.35 | 0.003 |
| Residuals | 8 | 1.003 | 0.125 | | |

| | Diff | Lower | Upper | Adjusted p |
|---|---|---|---|---|
| **Photoconversion** | | | | |
| Unconv vs Conv | 1.603 | 1.132 | 2.075 | <0.001 |
| **Tissue depth** | | | | |
| Surface vs Deep | 0.741 | 0.269 | 1.212 | 0.007 |
| **Photoconv:depth** | | | | |
| Unconv.Deep vs Conv.Deep | 0.727 | –0.199 | 1.653 | 0.132 |
| Conv.Surface vs Conv.Deep | –0.135 | –1.061 | 0.791 | 0.964 |
| Unconv.Surface vs Conv.Deep | 2.344 | 1.418 | 3.270 | <0.001 |
| Conv.Surface vs Unconv.Deep | –0.862 | –1.788 | 0.064 | 0.068 |
| Unconv.Surface vs Unconv.Deep | 1.617 | 0.691 | 2.543 | 0.002 |
| Unconv.Surf vs Conv.Surf | 2.478 | 1.553 | 3.405 | <0.001 |

*Appendix 1—table 3 Continued on next page*

**Appendix 1—table 3.** Results of linear regression with Bonferroni adjustment for multiple testing for *Appendix 1—figure 1*.

Green (505–560 nm) fluorescence emission

| Depth | Slope | SE | t value | $R^2$ adjusted | p | Adjusted p |
|---|---|---|---|---|---|---|
| 0 | 0.00902626 | 9.03E-05 | 99.9875929 | 0.99950008 | 6.00E-08 | 6.00E-07 |
| 70 | 0.0087601 | 7.62E-05 | 114.967832 | 0.99962183 | 3.43E-08 | 3.43E-07 |
| 140 | 0.01156811 | 0.00021343 | 54.2017461 | 0.99830038 | 6.94E-07 | 6.94E-06 |
| 210 | 0.01619148 | 0.00017715 | 91.4024378 | 0.9994018 | 8.59E-08 | 8.59E-07 |
| 280 | 0.02494033 | 0.00019703 | 126.581943 | 0.99968803 | 2.34E-08 | 2.34E-07 |
| 350 | 0.03021069 | 0.00018083 | 167.069327 | 0.99982089 | 7.70E-09 | 7.70E-08 |
| 420 | 0.03351092 | 0.0003004 | 111.554686 | 0.99959834 | 3.87E-08 | 3.87E-07 |
| 490 | 0.033718 | 0.00022538 | 149.603347 | 0.99977664 | 1.20E-08 | 1.20E-07 |
| 560 | 0.02842214 | 0.00042339 | 67.1300277 | 0.99889146 | 2.95E-07 | 2.95E-06 |
| 630 | 0.01999956 | 0.00028305 | 70.6577889 | 0.9989993 | 2.40E-07 | 2.40E-06 |

Red (560–650 nm) fluorescence emission

| Depth | Slope | SE | t value | $R^2$ adjusted | p | Adjusted p |
|---|---|---|---|---|---|---|
| 0 | 0.01719483 | 8.20E-05 | 209.593244 | 0.99988619 | 3.11E-09 | 3.11E-08 |
| 70 | 0.01510699 | 6.92E-05 | 218.286085 | 0.99989507 | 2.64E-09 | 2.64E-08 |
| 140 | 0.01881416 | 0.00019939 | 94.3604722 | 0.9994387 | 7.56E-08 | 7.56E-07 |
| 210 | 0.0266481 | 8.29E-05 | 321.256566 | 0.99995155 | 5.63E-10 | 5.63E-09 |
| 280 | 0.04096949 | 0.00025024 | 163.718467 | 0.99981349 | 8.35E-09 | 8.35E-08 |
| 350 | 0.05367283 | 0.00046253 | 116.040816 | 0.99962879 | 3.31E-08 | 3.31E-07 |
| 420 | 0.07300642 | 0.00047553 | 153.525421 | 0.9997879 | 1.08E-08 | 1.08E-07 |
| 490 | 0.09251997 | 0.00093074 | 99.4042684 | 0.99949419 | 6.14E-08 | 6.14E-07 |
| 560 | 0.08291338 | 0.00118633 | 69.8904363 | 0.99897723 | 2.51E-07 | 2.51E-06 |
| 630 | 0.06235948 | 0.00053652 | 116.22921 | 0.99962999 | 3.29E-08 | 3.29E-07 |

**Appendix 1—table 4.** List of acclimation conditions, experimental treatments, and measurements performed on each species and morph used in the study.

| Species and morph | Acclimation | Experimental treatment | Measurements |
|---|---|---|---|
| | | None | $E_0$ profiles (white illumination) |
| | | Exposure to xenon arc lamp +UV filter for 6 hr localised to a 3 mm spot | $E_0$ profiles inside and outside treated area (white and blue illumination) |
| *M. cavernosa* | 40 days green light | Exposure to near-UV LED over the entire sample for 10 hr | $E_0$ profiles before and after treatment (white and blue illumination) |
| *Echinophyllia* sp. green morph | >6 months broad spectrum metal halide and 100 μmol photons m$^{-2}$ s$^{-1}$ | None | $E_0$ profiles (blue illumination) |
| *Echinophyllia* sp. red morph | >6 months broad spectrum metal halide and 100 μmol photons m$^{-2}$ s$^{-1}$ | None | $E_0$ profiles (blue illumination) |

*Appendix 1—table 4 Continued on next page*

*Appendix 1—table 4 Continued*

| Species and morph | Acclimation | Experimental treatment | Measurements |
|---|---|---|---|
| *P. damicornis* pink morph | >6 months broad spectrum metal halide and 100 μmol photons m$^{-2}$ s$^{-1}$ | Exposure to 400 μmol photons m$^{-2}$ s$^{-1}$ red light for 7 days followed by exposure to 100 μmol photons m$^{-2}$ s$^{-1}$ blue light for 20 days | $E_0$ profile and surface $E_0$ after treatment (white illumination) |
| *P. damicornis* brown morph | >6 months broad spectrum metal halide and 100 μmol photons m$^{-2}$ s$^{-1}$ | None | $E_0$ profile and surface $E_0$ after treatment (white illumination) |
| *M. foliosa* purple edge morph | >6 months broad spectrum metal halide and 200 μmol photons m$^{-2}$ s$^{-1}$ | None | Surface $E_0$ (white illumination) |
| *M. foliosa* brown morph | >6 months broad spectrum metal halide and 200 μmol photons m$^{-2}$ s$^{-1}$ | 65 days high nitrate/low phosphate | Surface $E_0$ after treatment (white illumination) |

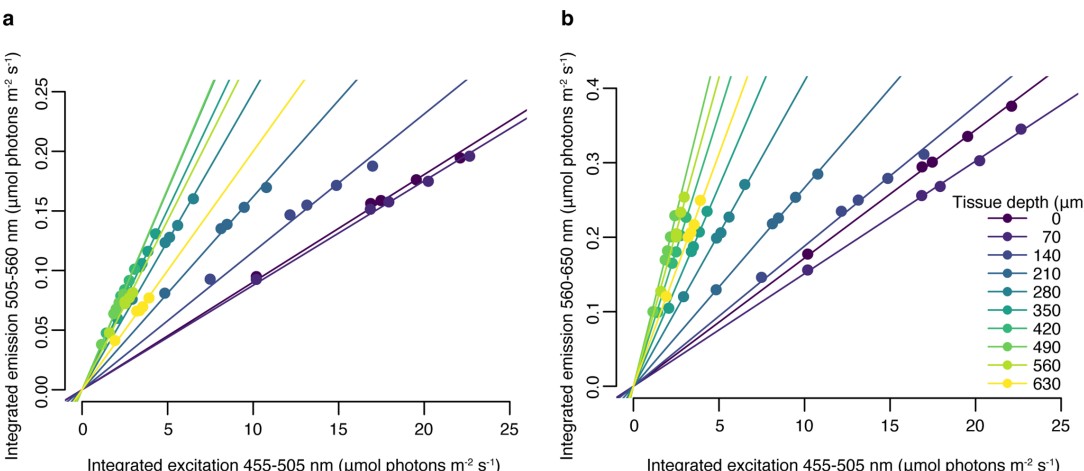

**Appendix 1—figure 1.** Relationship between blue (455–505 nm) excitation light photon irradiance measured *in hospite* and emitted green (505–560 nm), (**a**) or red (560–650 nm), (**b**) fluorescence at 10 tissue depths, obtained with a set of neutral density filters. Trendlines represent fitted linear regression (adjusted p<0.05, *Appendix 1— table 3*).

