## [Editor Report]

This work provides interesting insight into and hypotheses about the ecological roles of fluorescent proteins (like GFP) that many scientists use as tools for their experiments, but whose natural roles are little understood. The work provides evidence that they modify the light environment available to the organism (here a coral and its symbionts) to allow for photosynthesis even under low light conditions. Thus, the work provides an interesting mixture of insight to scientists of other disciplines to understand what they are working with and to ecologists/marine biologists to better understand light usage in the oceans. The paper is very well written.

---

## [Decision Letter]

**Decision letter after peer review:**

Thank you for submitting your article "Green fluorescent protein-like pigments optimize the internal light environment in symbiotic reef building corals" for consideration by *eLife*. Your article has been reviewed by 3 peer reviewers, and the evaluation has been overseen by a Reviewing Editor and Meredith Schuman as the Senior Editor. The following individual involved in the review of your submission has agreed to reveal their identity: Dominique Bourgeois (Reviewer #2).

Given that the main novelty of your work is within the light measurements, we ask you to submit the revised version in the category of "tools and resources". Please contact the *eLife* editorial office to modify the manuscript type in the system when you are ready to resubmit. Please take good care to clarify the novelty of your work now in the text of your manuscript.

Furthermore, we ask you to more clearly discuss the limitations of your study. It should be clear to the readers that you have not assessed how the light environment modified by the pcRFPs impacts on the photosynthetic activity of the symbionts and, therefore would contribute to the fitness of the coral.

This is particularly important given that it is still unclear to which extend the shift in the light spectrum can really contribute to the light usable for the symbionts.

As phrased by one of the reviewers: "What the authors show in Figure 4 is that the photon flux gained through penetrating fluorescence in the 560-650 band is less than a few photons and obviously decreases along with the decrease in the intensity of exciting light. In Figure 1b they show that the diffusion attenuation coefficient for the red (R) band is ~1 (mm-1) vs ~2 (mm-1) for the blue (B) band. Over a distance of 0.5 mm the R/B ratio in the incident light would increase by ~65%. However fluorescence emission is a small fraction of the exciting light intensity, generally <5%. In fact the authors state in the clarification letter that "Indeed, our data show that the contribution of pcRFPs to the holobiont light environment becomes the dominant source of orange-red light" and not "the dominant source of light". If the above is correct, B would still be the dominant source of energy for the symbiont, despite the strong attenuation by the tissue."

Please address these remarks and also discuss your measurement data in the context of the published knowledge on the Symbiodinium spectra- dependent photobiological responses (Wangpraseurt D et al., (2014) Spectral Effects on Symbiodinium Photobiology Studied with a Programmable Light Engine. PLoS ONE 9(11): e112809. https://doi.org/10.1371/journal.pone.0112809).

Please also carefully consider the following remarks, which will help significantly to improve the manuscript.

General comments:

The statistical significance of the presented data may not be sufficiently discussed. For example in Figure 1a, is the lower scalar irradiance by the converted coral, as compared to the unconverted coral, significant or not? If yes, why? In figure 2a (which has no error bars) is the apparent slight reduction in the 514/ 582 nm ratio at deep layers for the most photoconverted coral (dark red line) significant or not? If yes, why? In figure 3d and e (again no error bars), is the slight increase of green and red fluorescence emission in the deeper coral layers by unconverted PCFPs significant or not, and if yes why? In figure 4, to judge the significance of the quantitative evaluation, it would be important to show a comparison between the irradiance spectra of the Red Sea (figure 4A) with the lab-based spectra of figure S1.

Introduction

Lines 74-76: what changes at the coral polyp-scale are the Authors referring to? Please expand and add more recent references throughout this whole paragraph. Kramer et al., is missing the year.

Lines 95-96: light availability is certainly one of the factors that drive changes in growth morphology along a depth gradient, but not the only one. Moreover, a transition from branching forms in shallow-water to plate-like morphologies in deep-water is not so straightforward in all the species that thrive along steep depth gradients. Other possible factors that could contribute to changes in morphology are for instance hydrodinamism and therefore nutrients. Additionally, when investigating photosynthetic rates along a depth gradient, not just light but also temperature should be taken into account, as photosynthesis is a temperature-dependent process. These additional factors should be mentioned here.

Lines 100-101: Nutrient availability and therefore tissue composition (e.g., lipids) and thickness can change with depth. Could this affect tissue optical properties? How? Please elaborate.

Lines 107-108: Why only in some corals? Under which conditions do corals develop outstanding photosynthetic quantum efficiencies?

Lines 108-110: The Authors say that optical properties can change with ontogeny, thus throughout the lifetime of the coral. How do these properties change with ontogeny and how does this ontogenetic plasticity allow corals to occupy wider niches?

Lined 110-113: This last sentence seems unrelated to the previous one. What is the connection between changes in growth forms with ontogeny? Please rephrase.

Line 186: I would replace "environmental regulation" with "light regulation" since light was the only variable considered in this study.

In general, references in the Introduction are a bit outdated.

Results and Discussion

Line 197: what do the authors mean exactly with "all specimens"? In legend of Figure 1 the Authors say N = 3, three measurements for 2 fragments. Are these two fragments of the same colony? If this is the case, how can these measurements really account for the variability of a natural population, especially considering that the study was conducted on corals kept in aquaria for 10 years? If all measurements were conducted using fragments of the same colony you can't assess whether the differences you observe are statistically significant, because natural variability is eliminated. This should be clearly accounted for in the text without generalizing the results (see below, lines 233-237).

Lines 216-218: Has skeleton backscatter been measured at different depths in M. cavernosa? Could this species perhaps alter its morphology to increase the skeleton backscatter at greater depth, thus explaining why it thrives at mesophotic depths?

Lines 233-237: Authors should downscale this sentence and specify that this is what they found under these experimental conditions. This cannot be generalized to corals in shallow water environments. Moreover, the variability mentioned here by the Authors is not just given by physical properties but also by intra and inter specific variability.

Lines 301-302: How did the authors do this simulation? Please add details in the M and M section.

Line 304: Please correct photos with photons.

At lines 315/316 the authors mention possible fluorescence saturation at high light levels, but they don't discuss the involved mechanism (triplet state saturation?) nor give any estimation of numbers (excitation rates, intersystem crossing yield, triplet state lifetime, if this is the triplet state which is invoked) that could convince the reader.

Some of the figures present +s.d as dashed lines, and others +/- s.d as double dashed lines. Please keep consistency throughout the paper. Also some of the legends are not consistent with the plotted data in this regard.

Line 357, there is no spectrum in figure 5A.

Line 366: could the authors comment on enhancement of red/far-red light in figure 5C in the bleached + CP sample.

Figure 5C: what is the straight dashed line?

Line 408: attenuation coefficient should be Ko?

Line 414: What does N = 3 stand for? Fragments? Measurements?

Materials and methods

Line 433: The authors should add an explanation of the experimental design describing the experimental conditions, duration, and number of samples per species that were analyzed. The number of investigated samples per analysis is not clear.

Line 434: the aquarium setup is not described. Were the corals fed? How often and with what? What was the water temperature and salinity? How often were these parameters monitored? This is where you should describe the experimental design and why you chose to use these four species. Add information (either here or in the Introduction section) on the ecology of these species that can be useful to understand why you chose them as your study organisms.

Line 448-449: please specify the tissue thicknesses for all the investigated species.

Line 451: please explain why the microsensor was inserted at an angle of 45{degree sign} relative to the incident light.

Line 454: Why did you measure the spatial distribution of emitted fluorescence only for these two species? Please explain.

Lines 502-503: how many measurements on how many samples?

Lines 509-510: did you use a separate control aquarium? With how many specimens? If you used only 1 aquarium per treatment, did you take into account the "tank effect"?

Line 519: why did you bleach M. foliosa by phosphate starvation and didn't use the same bleaching method used for P. damicornis?

References

The reference list needs to be double checked (e.g., some references are incomplete) and page numbers included.

Line 582: pages are missing after volume.

Line 644: volume and page numbers missing.

Line 646-648: double-check the format of this reference, not sure its correct.

Supplementary Material

Figure S3: the red morph does not seem to exhibit any green fluorescence (panel b), yet there is a strange baseline that seemed to rise sharply below 500 nm. Please discuss this baseline.

Figure S5: why not showing the saturation behavior at higher integrated excitations? Also please specify that the reported integrated excitations correspond to those measured at each depth, and not the impinging integrated excitation at the coral surface (if correct).

Please also explain why there is more fluorescence per excitation photon at higher depth, either in the green or in the red spectral regions.

Table S1: please provide table legend. What is adjusted-P, which in fact is just P/10?

---

## [Author Response]

The reviewers have discussed their reviews with one another, and the Reviewing Editor has drafted this to help you prepare a revised submission.Given that the main novelty of your work is within the light measurements, we ask you to submit the revised version in the category of "tools and resources". Please contact the eLife editorial office to modify the manuscript type in the system when you are ready to resubmit. Please take good care to clarify the novelty of your work now in the text of your manuscript.

We thank the Reviewers and Editor for the positive evaluation of our manuscript and for the opportunity to provide a revised version. We have now included clarifications of the novelty of our work throughout the text, particularly with respect to Smith et al., 2013 and 2017 as previously suggested by the Reviewers and discussed with the Editor. The following paragraphs have been edited to follow this suggestion:

Introduction (lines 213-226)

“While compelling, the vast majority of evidence provided so far to support the role of GFP-like proteins in modulating the coral photosymbiont light environment during photoacclimation is indirect. This includes light measurements taken outside of the coral tissue (Smith et al., 2017, 2013), optical simulations (Lyndby et al., 2016), and experiments performed on coral-related animals (e.g. corallimorpharians) and isolated symbionts (Smith et al., 2017, 2013). So far, quantitative in vivo and in situ measurements of the symbiont light environment in response to light-driven changes in host pigment composition have been lacking. Here, we used advanced optical sensor technology to directly quantify how light-driven regulation of GFP-like proteins affects the intra-tissue light microenvironment of depth-generalist and shallow water corals. We focused on the effects of (i) UV-driven photoconversion in corals containing pcRFPs, and (ii) blue light-driven upregulation of inducible CPs during bleaching and in active growth zones. Our study provides an essential step towards understanding the role of these pigments in optimizing the functioning of the coral-algal symbiosis under extreme light regimes”

Discussion (lines 299-310)

“Using isolated symbionts and the sea anemone Discosoma as model systems for the symbiosis between cnidarian and Symbiodiniaceae, Smith et al., (2017) showed that orange light has a higher potential to stimulate photosynthesis in symbiont cells found deeper in the tissue as compared to blue-green light. This effect is analogous to what is observed in plant leaves, where enhanced illumination with green light increases photosynthesis in deeper cell layers (de Mooij et al., 2016; Vogelmann and Evans, 2002). Therefore, based on this indirect evidence it was proposed that orange fluorescence emission by pcRFPs could provide an energetic advantage for symbiotic corals in greater water depths dominated by blue light (Smith et al., 2017). Indeed, colour morphs with pcRFPs survived longer than non-pigmented conspecifics in simulated deep water light environments (Smith et al., 2017).

In this study, we aimed to directly measure the impact of pcRFPs on the in hospite light environment experienced by algal symbionts in coral tissue.”

Discussion (lines 424-430)

“Screening by CPs via absorption was proposed as a photoprotective mechanisms in shallow water corals by Smith et al., (2013), who used external reflectance measurements as well as measurements on freshly isolated symbionts in seawater to provide indirect support for their hypothesis. Our data thus provide the first direct evidence that coral CPs do indeed reduce light penetration within their absorption spectrum, dramatically altering the in hospite light environment of coral symbionts.”

Furthermore, we ask you to more clearly discuss the limitations of your study. It should be clear to the readers that you have not assessed how the light environment modified by the pcRFPs impacts on the photosynthetic activity of the symbionts and, therefore would contribute to the fitness of the coral.This is particularly important given that it is still unclear to which extend the shift in the light spectrum can really contribute to the light usable for the symbionts.As phrased by one of the reviewers: "What the authors show in Figure 4 is that the photon flux gained through penetrating fluorescence in the 560-650 band is less than a few photons and obviously decreases along with the decrease in the intensity of exciting light. In Figure 1b they show that the diffusion attenuation coefficient for the red (R) band is ~1 (mm-1) vs ~2 (mm-1) for the blue (B) band. Over a distance of 0.5 mm the R/B ratio in the incident light would increase by ~65%. However fluorescence emission is a small fraction of the exciting light intensity, generally <5%. In fact the authors state in the clarification letter that "Indeed, our data show that the contribution of pcRFPs to the holobiont light environment becomes the dominant source of orange-red light" and not "the dominant source of light". If the above is correct, B would still be the dominant source of energy for the symbiont, despite the strong attenuation by the tissue."

We fully agree with the Reviewer’s comment and interpretation. Indeed, pcRFPs only provide higher illumination in the orange-red waveband at mesophotic depths compared to the ambient/incident light. We state this in line 336-340, which reads:

“As a consequence, the pcRFP pool between 50 and 80 m depth can effectively double the scalar irradiance in the 560-650 nm waveband compared to corals without pcRFPs (Figure 4b). On mesophotic reefs beyond 80 m water depth, simulated fluorescence emission by the pcRFP becomes the dominant source of 560-650 nm light for this M. cavernosa specimen.”

We discuss this further in another paragraph, which we have edited to more clearly reflect the Reviewer’s point and which now reads (lines 355-366):

“Even though this constitutes a dramatic spectral alteration in relative terms, it is important to note that (1) the absolute number of photons generated by pcRFP fluorescence is small compared to what is provided by ambient light in shallow waters and (2) the in hospite irradiance is still dominated by the blue-green spectrum, which should be regarded as the main source of irradiance for coral symbionts in the mesophotic zone. Thus, under conditions of high solar radiation, the benefit of pcRFP fluorescence for photosynthesis may be negligible compared to the potential energetic investments related to maintaining a dense pcRFP pigment pool (Leutenegger et al., 2007; Oswald et al., 2007; Quick et al., 2018). However, under strong light-limitation (<40 μmol photons m^-2^ s^-1^) pcRFP-driven photosynthesis might be energetically advantageous, as suggested by the increased long term survival of pcRFP-containing coral morphs compared to their brown conspecifics under mesophotic light regimes (Smith et al., 2017).”

Please address these remarks and also discuss your measurement data in the context of the published knowledge on the Symbiodinium spectra- dependent photobiological responses (Wangpraseurt D et al., (2014) Spectral Effects on Symbiodinium Photobiology Studied with a Programmable Light Engine. PLoS ONE 9(11): e112809. https://doi.org/10.1371/journal.pone.0112809).

We have now expanded our conclusion section to include what is currently known about the response of Symbiodiniaceae to spectral light in hospite, and how this information is relevant to our study. The section reads (lines 470-483):

“To date, very few studies have explored the effects of a red-shifted spectrum on symbiont photobiology in hospite (Lichtenberg et al., 2016; Szabó et al., 2014; Wangpraseurt et al., 2014c). The available data suggests that red illumination can support similar levels of photosynthetic production in deeper tissue areas compared to shallow ones, despite reduced light availability deeper in the tissue (Lichtenberg et al., 2016). Similar rates of oxygen evolution were also observed in hospite for symbionts exposed to either blue-shifted or red-shifted broad illumination, however under low light conditions red-shifted illumination resulted in lower electron transport rates (Wangpraseurt et al., 2014c). Finally, in cultured coral symbionts, the absorption cross-section of PSII is over 3x higher in the blue region compared to the orange-red, however this difference becomes less pronounced when the symbionts are studied in hospite (Szabó et al., 2014). A more complete understanding of how coral symbionts acclimate to and utilise orange-red light in hospite is clearly needed in order to understand how spectral tuning of this waveband by host pigments impacts holobiont photosynthesis.”

Please also carefully consider the following remarks, which will help significantly to improve the manuscript.General comments:The statistical significance of the presented data may not be sufficiently discussed. For example in Figure 1a, is the lower scalar irradiance by the converted coral, as compared to the unconverted coral, significant or not? If yes, why? In figure 2a (which has no error bars) is the apparent slight reduction in the 514/ 582 nm ratio at deep layers for the most photoconverted coral (dark red line) significant or not? If yes, why? In figure 3d and e (again no error bars), is the slight increase of green and red fluorescence emission in the deeper coral layers by unconverted PCFPs significant or not, and if yes why? In figure 4, to judge the significance of the quantitative evaluation, it would be important to show a comparison between the irradiance spectra of the Red Sea (figure 4A) with the lab-based spectra of figure S1.

We have included the results of further statistical analyses in the text, the figures and the appendix.

Figure 1a. The difference between the two morphs is not significant at any of the wavelengths tested. We have included this information in the text, which now reads (lines 232-235):

“This enhancement was detected in both the unconverted and converted tissue areas measured, with no significant differences between the two in any of the wavebands tested (t-test with Bonferroni adjustment, adjusted p>0.05. Figure 1a, Appendix 1—table 1).”

The new Appendix 1—table 1 presents the results of this analysis.

Figure 2a

Individual regions on the coral surface presented slight differences in the surface 514:582 nm ratio and, as a consequence, each measurement had a unique spectral distribution (Figure 2c). Therefore, we chose to present the data as individual profiles, in order to better highlight these fine spectral variations. Alongside this display, we have now also grouped the data into “converted” and “unconverted” (new Figure 2a boxplots) and provided the results of statistical analysis in the text, which now reads (lines 283-289):

“This ratio was significantly higher in unconverted corals, compared to those that had been exposed to UV light (Figure 2—figure supplement 2) for any duration (partially converted or converted corals, Two-way ANOVA followed by Tukey’s post-hoc test, adjusted p <0.01. Figure 2, Appendix 1—table 2). The ratio showed a significant decrease from surface to 350 μm tissue depth only for unconverted corals (adjusted p <0.01), and no significant differences were detected between the two groups at 350 μm tissue depth (adjusted p >0.05).”

Results of this analysis are provided in the new Appendix 1—table 2.

Figure 3d and 3e

As explained above, the fluorescence emission at different tissue depths is displayed as individual profiles in order to capture the spectral variability between different areas on the coral surface. Naturally, these areas are variable in terms of tissue thickness. As a consequence, the recorded profiles vary in their depth coverage and prevent consistent averaging over the length for statistical purposes. For this reason, we have not attempted statistical comparison between these profiles, and have only used the data in this figure as a quantification of light generated via fluorescence emission. The edited text now reads (lines 314-320):

“Fluorescence emission by the pcRFP pool in the most converted state provided an amount of orange-red photons that ranged between 1% (close to the skeleton) and 5% (at the coral surface) of the incident blue irradiance, with intermediate values of around 3% at a tissue depth of 400 μm (Figure 3e). This direct measurement demonstrates that orange-red fluorescence from host pigments penetrates well in coral tissue. Furthermore, these values show that ectodermal pcRFPs can improve the illumination of deeper tissue layers exposed to the narrow mesophotic light spectrum.”

Figure 4

We have now overlayed the 80 m spectrum from the Red Sea on Figure 2—figure supplement 1c. We comment on the spectral comparison between this spectrum and the measuring light source in the text, which now reads (lines 277-282):

“The peak emission wavelength of this light source matches (± 3 nm) the irradiance maximum measured at 80 m depth in the Red Sea (Eyal et al., 2015), and over 50% of irradiance at 80m falls within the 455-505 nm band (Figure 2—figure supplement 1). Therefore, while not a perfect match, this light source is spectrally more representative of mesophotic light conditions compared to the broad-spectrum LED (Figure 2—figure supplement 1).”

IntroductionLines 74-76: what changes at the coral polyp-scale are the Authors referring to? Please expand and add more recent references throughout this whole paragraph. Kramer et al., is missing the year.

We are referring to behavioural changes at the polyp scale, now rephrased (lines 74-78):

“Some of these mechanisms occur at the symbiont cell level, while others involve changes in the coral colony morphology and polyp behaviour (Hoogenboom et al., 2008; Kahng et al., 2019; Kaniewska et al., 2014; Kaniewska and Sampayo, 2022; Kramer et al., 2021; Levy et al., 2003; Todd, 2008).”

Following this and other comments, we have now expanded the entire section to be more comprehensive of both symbiont-level and host-level mechanism. The edited text reads (lines 78-140):

“Coral symbionts can modify their light harvesting and energy quenching over timescales ranging from seconds to months. For example, non-photochemical quenching allows rapid dissipation of excess irradiance in the photosynthetic apparatus of symbionts, preventing photodamage in high light environments (Einbinder et al., 2016; Gorbunov et al., 2001; Hoegh-Guldberg and Jones, 1999).

Coral symbionts associated with deeper water colonies have been frequently found to differ from their shallow water counterparts in terms of photosystem II photochemical efficiency, maximum electron transport rates, and the organization of light harvesting antennae (Einbinder et al., 2016; Hennige et al., 2008; Lohr et al., 2019).

Photosynthetic pigment concentrations and symbiont cell numbers are dynamic within coral colonies, enabling them to acclimate to changes in irradiance; specifically, corals can increase their symbiont numbers as well as the per cell photosynthetic pigment content in response to a reduction in light levels, and vice versa (Eyal et al., 2019; Falkowski and Dubinsky, 1981; Martinez et al., 2020; Tamir et al., 2020; Titlyanov et al., 2001). Symbiont community composition within colonies is also known to vary along light gradients, although the plasticity of this association and the extent to which it can contribute to photoacclimation appears to be species-specific (Chan et al., 2009; Einbinder et al., 2016; Kaniewska and Sampayo, 2022; Lesser et al., 2010; Martinez et al., 2020; Nir et al., 2011; Winters et al., 2009). Additionally, the interplay between light and temperature is likely to play a role in shaping these communities (Kahng et al., 2019; Kahng and Kelley, 2007), as the low temperatures sometimes encountered on mesophotic reefs affect the photosynthetic rates of symbiont species differently (Grégoire et al., 2017).

Coral host morphology is an important factor controlling the irradiance available for symbiont photosynthesis (Einbinder et al., 2009; Helmuth et al., 1997; Kahng et al., 2019, 2012; Kaniewska et al., 2011; Kaniewska and Sampayo, 2022; Nir et al., 2011; Ralph et al., 2002; Wangpraseurt et al., 2014b). Coral colonies from low light environments maximize interception with the downwelling irradiance and assume plate-like morphologies that are more conducive to light collection, while shallow water colonies assume self-shading morphologies, such as vertical plates or branches (Anthony et al., 2005; Kaniewska et al., 2011). Morphological adaptations make some species depth-specialists, such as the mesophotic coral Leptoseris which can host photosynthetic symbionts down to depths of well over 100 m (Kahng et al., 2020; Rouzé et al., 2021). Depth generalist corals on the other hand, such as Montastraea cavernosa in the Caribbean, thrive across light gradients of several orders of magnitude (Lesser et al., 2010). These corals often assume different morphologies in different light environments, allowing a single species to occupy a wider environmental niche; for example, on mesophotic reefs, the depth-generalist corals Paramontastrea peresi and Porites lobata have growth forms that enhance light exposure and their skeletons have higher reflectivity compared to their shallow water conspecifics, resulting in higher absorption by the symbionts (Kramer et al., 2020). Such morphological differences are partially due to phenotypic plasticity, as shown by transplantation experiments across depth gradients (Muko et al., 2000; Willis, 1985). The relationship between light and colony morphology is not always predictable (Doszpot et al., 2019) as other environmental variables such as hydrodynamics also affect colony morphology (Soto et al., 2019).

Light availability to the coral symbionts depends on the optical properties of the surrounding host tissue and the underlying skeleton (Marcelino et al., 2013; Wangpraseurt et al., 2019, 2012). Coral tissue and skeleton scatter incident light with a strength and directionality that are highly variable across species and even between different structures within a single colony (Enríquez et al., 2017; Marcelino et al., 2013; Wangpraseurt et al., 2019, 2016a). The interplay between tissue and skeleton scattering and reflection and algal absorption modulate coral photosynthesis (Brodersen et al., 2014; Marcelino et al., 2013; Scheufen et al., 2017; Wangpraseurt et al., 2019, 2016a). This can lead to outstanding photosynthetic quantum efficiencies that are close to the theoretical maximum, at least for the coral species for which this parameter has been investigated (Brodersen et al., 2014).

Aside from variability in skeletal morphology, tissue-level changes such as contraction and expansion are known to affect light penetration (Wangpraseurt et al., 2017b, 2014a). The thickness and composition of the tissue can be affected by a number of variables, including genotype, nutritional status, seasonality and environmental history (Harland et al., 1992; Jones et al., 2020; Leinbach et al., 2021; Lough and Barnes, 2000; Rocker et al., 2019). Changes in tissue composition (e.g. lipids, collagen, proteins) will affect the light scattering properties of the bulk tissue (Jacques, 2013), although detailed investigations on coral optical properties in relation to tissue composition and nutrient status remain to be performed (Lyndby et al., 2019).”

We have also revisited our reference list and added more recent ones as suggested. These include Kahng et al. 2019, Eyal et al., 2019, Kaniewska and Sampayo 2022, Einbinder et al., 2016, Lohr et al. 2019, Martinez et al., 2020, Grégoire et al., 2017, Doszpot et al., 2019, Soto et al. 2019, Wangpraseurt et al. 2019, Jones et al. 2020, Leinbach et al. 2021, Rocker et al. 2019.

Lines 95-96: light availability is certainly one of the factors that drive changes in growth morphology along a depth gradient, but not the only one. Moreover, a transition from branching forms in shallow-water to plate-like morphologies in deep-water is not so straightforward in all the species that thrive along steep depth gradients. Other possible factors that could contribute to changes in morphology are for instance hydrodinamism and therefore nutrients. Additionally, when investigating photosynthetic rates along a depth gradient, not just light but also temperature should be taken into account, as photosynthesis is a temperature-dependent process. These additional factors should be mentioned here.

We have now mentioned these additional factors in the text, which now reads:

Lines 116-121

“Such morphological differences are partially due to phenotypic plasticity, as shown by transplantation experiments across depth gradients (Muko et al., 2000; Willis, 1985). The relationship between light and colony morphology is not always predictable (Doszpot et al., 2019) as other environmental variables such as hydrodynamics also affect colony morphology (Soto et al., 2019).”

Lines 91-99

“Symbiont community composition within colonies is also known to vary along light gradients, although the plasticity of this association and the extent to which it can contribute to photoacclimation appears to be species-specific (Chan et al., 2009; Einbinder et al., 2016; Kaniewska and Sampayo, 2022; Lesser et al., 2010; Martinez et al., 2020; Nir et al., 2011; Winters et al., 2009). Additionally, the interplay between light and temperature is likely to play a role in shaping these communities (Kahng et al., 2019; Kahng and Kelley, 2007), as the low temperatures sometimes encountered on mesophotic reefs affect the photosynthetic rates of symbiont species differently (Grégoire et al., 2017).”

Lines 100-101: Nutrient availability and therefore tissue composition (e.g., lipids) and thickness can change with depth. Could this affect tissue optical properties? How? Please elaborate.

We have now discussed this in the text, which now reads (lines 132-140):

“Aside from variability in skeletal morphology, tissue-level changes such as contraction and expansion are known to affect light penetration (Wangpraseurt et al., 2017b, 2014a). The thickness and composition of the tissue can be affected by a number of variables, including genotype, nutritional status, seasonality and environmental history (Harland et al., 1992; Jones et al., 2020; Leinbach et al., 2021; Lough and Barnes, 2000; Rocker et al., 2019). Changes in tissue composition (e.g. lipids, collagen, proteins) will affect the light scattering properties of the bulk tissue (Jacques, 2013), although detailed investigations on coral optical properties in relation to tissue composition and nutrient status remain to be performed (Lyndby et al., 2019).”

Lines 107-108: Why only in some corals? Under which conditions do corals develop outstanding photosynthetic quantum efficiencies?

The study we refer to deals primarily with 3D light distribution, propagation and absorption, and how these parameters affect quantum efficiency in corals. However, many other factors may affect quantum efficiency, and importantly this parameter has only been investigated in a limited number of coral species. Therefore we refrained from generalizing – we have now clarified this in the text, which reads (lines 129-131):

“This can lead to outstanding photosynthetic quantum efficiencies that are close to the theoretical maximum, at least for the coral species for which this parameter has been investigated (Brodersen et al., 2014).”

Lines 108-110: The Authors say that optical properties can change with ontogeny, thus throughout the lifetime of the coral. How do these properties change with ontogeny and how does this ontogenetic plasticity allow corals to occupy wider niches?Lined 110-113: This last sentence seems unrelated to the previous one. What is the connection between changes in growth forms with ontogeny? Please rephrase.

We used “ontogeny” when referring to the growth history of the colony – perhaps not the best choice, considering that the early life stages are not monitored. Thank you for pointing this out, we have now replaced it with “phenotypic plasticity” and have clarified the role of this plasticity in the text, which now reads (lines 109-118):

“Depth generalist corals on the other hand, such as Montastraea cavernosa in the Caribbean, thrive across light gradients of several orders of magnitude (Lesser et al., 2010). These corals often assume different morphologies in different light environments, allowing a single species to occupy a wider environmental niche; for example, on mesophotic reefs, the depth-generalist corals Paramontastrea peresi and Porites lobata have growth forms that enhance light exposure and their skeletons have higher reflectivity compared to their shallow water conspecifics, resulting in higher absorption by the symbionts (Kramer et al., 2020). Such morphological differences are partially due to phenotypic plasticity, as shown by transplantation experiments across depth gradients (Muko et al., 2000; Willis, 1985).”

Line 186: I would replace "environmental regulation" with "light regulation" since light was the only variable considered in this study.

Replaced, the sentence now reads (line 220-222):

“Here, we used advanced optical sensor technology to directly quantify how light-driven regulation of GFP-like proteins affects the intra-tissue light microenvironment of depth-generalist and shallow water corals”

In general, references in the Introduction are a bit outdated.

We have now revisited our reference list and added more recent ones as suggested. These include Kahng et al. 2019, Eyal et al. 2019, Kaniewska and Sampayo 2022, Einbinder et al. 2016, Lohr et al. 2019, Martinez et al. 2020, Grégoire et al. 2017, Doszpot et al. 2019, Soto et al. 2019, Wangpraseurt et al. 2019, Jones et al. 2020, Leinbach et al. 2021, Rocker et al. 2019.

Results and DiscussionLine 197: what do the authors mean exactly with "all specimens"? In legend of Figure 1 the Authors say N = 3, three measurements for 2 fragments. Are these two fragments of the same colony? If this is the case, how can these measurements really account for the variability of a natural population, especially considering that the study was conducted on corals kept in aquaria for 10 years? If all measurements were conducted using fragments of the same colony you can't assess whether the differences you observe are statistically significant, because natural variability is eliminated. This should be clearly accounted for in the text without generalizing the results (see below, lines 233-237).

We have added further details on the type and number of replicates in the Methods section, which now reads (lines 489-495):

“Coral colonies of M. cavernosa (one genet, three ramets), Echinophyllia sp. (two genets, one ramet of each), Pocillopora damicornis (two genets, one ramet) and Montipora foliosa (two genets, one ramet of each) were asexually propagated for >10 years in the experimental coral aquarium facility of the Coral Reef Laboratory, National Oceanography Centre, Southampton, UK (D’Angelo and Wiedenmann 2012b). Each sample represents an autonomous colony with an individual, long-term growth history. Therefore, while genetically identical, these samples can be considered true biological replicates.”

We fully appreciate the importance of variability in natural populations, and acknowledge that the use of clonal replicates implies that we cannot generalise our results to all individuals of our test species. However, we also believe that experiments performed on clonal replicates are an essential step in mechanistic studies, for the very reason that they allow us to control for genetic differences between test subjects and give us high confidence that the response measured is indeed the result of our experimental treatment. We have added a paragraph to explain our choice and to acknowledge the limitations of our approach in the text, which now reads (lines 381-392):

“All these variables can affect optical properties and photosynthetic performance, and potentially contribute to the success of this species across steep light gradients. In this study, we controlled for these variables by choosing to perform our measurements on clonal replicates from long-term aquarium culture. This approach allows us to confidently attribute the measured changes in internal light environment to the changes in the pcRFP complement that we induced experimentally. Further studies are welcome to analyse if and how other variables may interact with pcRFPs functions to shape the symbiont light environment along natural depth gradients. Finally, intraspecific colour polymorphism of reef corals is frequently driven by different tissue concentrations of GFP-like proteins including CPs and pcRFPs (Gittins et al., 2015; Quick et al., 2018; Smith et al., 2017). Accordingly, representatives of different colour morphs can be expected to show deviating responses.”

We have also clarified this in the Methods section, which now reads (lines 498-502):

“Additionally, their very long-term growth and propagation in our stable aquarium conditions ensured that these individuals were fully acclimated to the pre-experimental conditions, thus allowing to ascribe any observed patterns exclusively to the manipulated parameters – something that would not be straightforward for field collected or recently acquired colonies.”

Lines 216-218: Has skeleton backscatter been measured at different depths in M. cavernosa? Could this species perhaps alter its morphology to increase the skeleton backscatter at greater depth, thus explaining why it thrives at mesophotic depths?

This is a great suggestion, which unfortunately has not yet been properly investigated for this particular species. Upon further reading, we found one study that reported a subtle difference in the corallite structure for M. cavernosa collected from different depths. We have now included this information, along with other potential sources of variability in optical properties, in the text which now reads (lines 376-383):

“For M. cavernosa, one study reported a small difference in skeletal microstructure between colonies collected from different depths (Beltrán-Torres and Carricart-Ganivet, 1993). Another study revealed a shift towards heterotrophy and a change in symbiont community composition for the same species along a shallow to mesophotic gradient (Lesser et al., 2010). Genetic structuring of the M. cavernosa population with depth has also been reported (Brazeau et al., 2013). All these variables can affect optical properties and photosynthetic performance, and potentially contribute to the success of this species across steep light gradients.”

Lines 233-237: Authors should downscale this sentence and specify that this is what they found under these experimental conditions. This cannot be generalized to corals in shallow water environments. Moreover, the variability mentioned here by the Authors is not just given by physical properties but also by intra and inter specific variability.

We agree with this statement and have rephrased the sentence (lines 263-266):

“Therefore it appears that, for this pcRFP-containing coral specimen, host pigment-derived enhancement of light would be a minor component for the overall tissue light microclimate in a shallow water environment,”

We have also added some considerations about intraspecific variability at lines 375-383:

“It is important to note that host pigments are only one of many variables that influence the optical properties of coral colonies along a depth gradient. For M. cavernosa, one study reported differences in skeletal microstructure between colonies collected from different depths (Beltrán-Torres and Carricart-Ganivet, 1993). Another study revealed a shift towards heterotrophy and a change in symbiont community composition for the same species along a shallow to mesophotic gradient (Lesser et al., 2010). Genetic structuring of the M. cavernosa population with depth has also been reported (Brazeau et al., 2013). All these variables can potentially affect bio-optical properties and/or photosynthesis, and thus affect the success of this species across steep light gradients.”

Lines 301-302: How did the authors do this simulation? Please add details in the M and M section.

Added, the section now reads (lines 624-628):

“These values were used to calculate estimates of red fluorescence emission for the converted M. cavernosa at different water depths, by multiplying the integral of the 455-505 nm field radiance by the percentage of incident light re-emitted as 560-650 nm fluorescence shown in Figure 3e.”

Line 304: Please correct photos with photons.

Edited, thank you!

At lines 315/316 the authors mention possible fluorescence saturation at high light levels, but they don't discuss the involved mechanism (triplet state saturation?) nor give any estimation of numbers (excitation rates, intersystem crossing yield, triplet state lifetime, if this is the triplet state which is invoked) that could convince the reader.

We meant to indicate the possibility that the incident light is sufficient to excite all chromophores at the same time. Under this hypothetical scenario, additional illumination would not result in additional fluorescence emission. We have reworded the paragraph to make this clearer (lines 345-348):

“We note that our analysis could lead to overestimation of predicted red fluorescence at shallow depths, where high light levels may theoretically excite the entire chromophore population thus resulting in a non-linear relationship between incident light and fluorescence emission.”

Some of the figures present +s.d as dashed lines, and others +/- s.d as double dashed lines. Please keep consistency throughout the paper. Also some of the legends are not consistent with the plotted data in this regard.

Thank you for pointing this out, we have now edited Figure 1 and Figure 5 to only display + s.d., and have edited the figure legends accordingly

Line 357, there is no spectrum in figure 5A.

Edited, we meant to refer to Figure 5b. The text now reads (lines 401-402):

“The bare white skeleton of P. damicornis enhanced spectral scalar irradiance by 140-150% across the entire visible spectrum (Figure 5b).”

Line 366: could the authors comment on enhancement of red/far-red light in figure 5C in the bleached + CP sample.

Added, the sentence now reads (lines 411-413):

“conversely, blue and red light are enhanced in bleached pink tissue, while orange light is attenuated (Figure 5c).”

Figure 5C: what is the straight dashed line?

The dashed line was supposed to show K0=0 to aid with visualization of positive vs negative attenuation coefficient. It had accidentally been moved during figure formatting. We have now removed it so it would not create confusion with the standard deviation lines.

Line 408: attenuation coefficient should be Ko?

Edited, now reads (line 1033):

“(c) Spectral attenuation coefficient (K_0_(λ)) of scalar irradiance”

Line 414: What does N = 3 stand for? Fragments? Measurements?

Edited, now reads (lines 1038-1039):

“In b and e, mean (solid lines) + s.d. (dashed lines), n=3 measurements.”

Materials and methodsLine 433: The authors should add an explanation of the experimental design describing the experimental conditions, duration, and number of samples per species that were analyzed. The number of investigated samples per analysis is not clear.

We have now included the number and nature of replicates in the text, which now reads (lines 489-491):

“Coral colonies of M. cavernosa (one genet, three ramets), Echinophyllia sp. (two genets, one ramet of each), Pocillopora damicornis (two genets, one ramet) and Montipora foliosa (two genets, one ramet of each)”

We have also condensed all experimental details into the new Appendix 1—table 4.

Line 434: the aquarium setup is not described. Were the corals fed? How often and with what? What was the water temperature and salinity? How often were these parameters monitored? This is where you should describe the experimental design and why you chose to use these four species. Add information (either here or in the Introduction section) on the ecology of these species that can be useful to understand why you chose them as your study organisms.

We have added more details about the aquarium set up in this section (lines 503-509):

“All corals were acclimated for >6 months to the light conditions specified in Appendix 1—table 4, at a temperature of ~25°C, salinity 31, and were fed twice weekly with rotifers. Nutrient levels were maintained around ~6.5μM nitrate and ~0.3 μM phosphate (D’Angelo and Wiedenmann 2012b; Wiedenmann et al. 2013) unless otherwise specified in the text and in Appendix 1—table 4. A detailed description of the aquarium set up including flow rates, volumes, co-cultured species and monitoring of environmental parameters is presented in (D’Angelo and Wiedenmann 2012b).”

For brevity, we have referred the reader to the extensive description of the experimental mesocosm presented in D’Angelo and Wiedenmann 2012. More details on the experimental set up are now presented in Table S4. We have also added a paragraph to explain our species selection, which reads (lines 495-502):

“These species and colour morphs were selected for their established production of pcRFPs (M. cavernosa and Echinophyllia sp.) or photoprotective CPs (P. damicornis and M. foliosa) (Bollati et al., 2020; D’Angelo et al., 2012; Eyal et al., 2015; Oswald et al., 2007). Additionally, their very long-term growth and propagation in our stable aquarium conditions ensured that these individuals were fully acclimated to the pre-experimental conditions, thus allowing to ascribe any observed patterns exclusively to the manipulated parameters – something that would not be straightforward for field collected or recently acquired colonies.”

Line 448-449: please specify the tissue thicknesses for all the investigated species.

Added, the text now reads (lines 518-520):

“The step size was adjusted according to the total coral tissue thickness with step sizes of 20 µm for P. damicornis (tissue thickness ~150-300 µm), 70 µm for M. cavernosa (tissue thickness 400-800 µm) and Echinophyllia (tissue thickness 900 µm).”

Line 451: please explain why the microsensor was inserted at an angle of 45{degree sign} relative to the incident light.

Specified, now reads (lines 524-526):

“This measurement geometry was chosen to minimise artificial shading of the sensor tip while still allowing sensor penetration to the full tissue depth (Rickelt et al., 2016; Wangpraseurt et al., 2012).”

Line 454: Why did you measure the spatial distribution of emitted fluorescence only for these two species? Please explain.

We performed measurements with blue excitation only for corals involved in the pcRFP part of the study, because wavelength conversion of blue light into orange light was the proposed mechanism through which these pigments function. Additionally, blue illumination is highly relevant to mesophotic corals due to the reduced light spectrum experienced in this environment. For the CP part of the study, blue excitation measurements were not performed because (1) CPs are commonly found in shallow water environments, where corals experience a broader spectrum and (2) CPs emit no or only negligible fluorescence upon blue light excitation.

Lines 502-503: how many measurements on how many samples?

We have now included this information in the text, which reads (580-582):

“we performed measurements on one healthy and one bleached individual each of P. damicornis and M. foliosa (3 replicate measurements per sample per condition).”

We have also included information about replication at the beginning of the Methods section (lines 489-493):

“Coral colonies of M. cavernosa (one genet, three ramets), Echinophyllia sp. (two genets, one ramet of each), Pocillopora damicornis (two genets, one ramet) and Montipora foliosa (two genets, one ramet of each) were asexually propagated for >10 years in the experimental coral aquarium facility of the Coral Reef Laboratory, National Oceanography Centre, Southampton, UK (D’Angelo and Wiedenmann 2012b).”

We have added more information about the type and number of replicates in the legend for Figure 5, which now reads (lines 1038-1039):

“In b and e, mean (solid lines) + s.d. (dashed lines), n=3 measurements.”

Lines 509-510: did you use a separate control aquarium? With how many specimens? If you used only 1 aquarium per treatment, did you take into account the "tank effect"?

The experiment was performed in the same flow through system where we keep the stock specimens, which we used as visual control. All water parameters are therefore the same for bleached vs unbleached, with the exception of illumination. We have clarified this in the text, which now reads (lines 588-590):

“Bleaching was confirmed by visual comparison with >10 untreated specimens, which were kept in a separate area of the same aquarium under broad illumination from a metal halide lamp.”

Line 519: why did you bleach M. foliosa by phosphate starvation and didn't use the same bleaching method used for P. damicornis?

We performed this study at the same time as a stress experiment where we bleached corals using phosphate starvation. As we only needed specimens with the bleached vs unbleached phenotype in order to compare optical properties, we used specimens from this experiment (bleaching conditions described in the text at lines 600-604 and the new Appendix 1—table 1) and did not deem it necessary to sacrifice more colonies in order to use a different bleaching method.

ReferencesThe reference list needs to be double checked (e.g., some references are incomplete) and page numbers included.Line 582: pages are missing after volume.Line 644: volume and page numbers missing.Line 646-648: double-check the format of this reference, not sure its correct.

Edited

Supplementary MaterialFigure S3: the red morph does not seem to exhibit any green fluorescence (panel b), yet there is a strange baseline that seemed to rise sharply below 500 nm. Please discuss this baseline.

This is the tail of the spectrum of a CFP, which is found in the red morph. We have stated this in the text at lines 575-577, which read:

“The green morph contained a non-inducible GFP, while the red morph contained a CFP and a highly converted pcRFP (green:red emission ratio <<1) (Alieva et al., 2008; Bollati et al., 2017; Smith et al., 2017).”

Figure S5: why not showing the saturation behavior at higher integrated excitations? Also please specify that the reported integrated excitations correspond to those measured at each depth, and not the impinging integrated excitation at the coral surface (if correct).Please also explain why there is more fluorescence per excitation photon at higher depth, either in the green or in the red spectral regions.

Unfortunately, due to limitation in our light source we were only able to perform the calibration at these specific intensities at the time. This is why we have explicitly highlighted in the text this potential limitation of the data for higher irradiances, and have only made conservative conclusions based on the part of the dataset that was within the range of this calibration.

The integrated excitation is indeed measured at each depth, the reviewer’s interpretation is correct. Now specified in the text, which reads (Appendix 1):

“We then calculated the integrated photon scalar irradiance in the incident range (455-505 nm) at each tissue depth”

And in the figure legend (Appendix 1—figure 1):

”Relationship between blue (455-505 nm) excitation light photon irradiance measured in hospite and emitted green (505-560 nm, **a**) or red (560-650 nm, **b**) fluorescence at 10 tissue depths”

The trend highlighted by the reviewer arises from the fact that the incident waveband is attenuated more strongly compared to the fluorescence emission wavebands, therefore it appears that there is proportionally more fluorescence per excitation photon at depth. We have provided the complete set of measurements at various depth to show that the linear relationship is robust, since it remains even under very different optical properties such as those encountered deep in the tissue.

Table S1: please provide table legend. What is adjusted-P, which in fact is just P/10?

Adjusted p is after Bonferroni correction for multiple testing. This corresponds to p divided by number of tests and we ran 10 tests, that is why adjusted P is p/10. We have described the p adjustment in the Appendx 1 section “Incident intensity calibration (Supplementary Methods)”, which reads:

“A linear regression was fitted for each tissue depth and each chromophore, with intercept set to zero (Appendix 1—figure 1); p-values were corrected for multiple testing using the Bonferroni adjustment.”

We have also included this in the Appendix 1—table 3 legend, which now reads:

“Results of linear regression with Bonferroni adjustment for multiple testing for Appendix 1—figure 1.”